# A single N-terminal phosphomimic disrupts TDP-43 polymerization, phase separation, and RNA splicing

Ailin Wang[1,†], Alexander E Conicella[2,†], Hermann Broder Schmidt[3], Erik W Martin[4], Shannon N Rhoads[5], Ashley N Reeb[6], Amanda Nourse[4], Daniel Ramirez Montero[1], Veronica H Ryan[7], Rajat Rohatgi[3,8], Frank Shewmaker[5], Mandar T Naik[1], Tanja Mittag[4] (ID), Yuna M Ayala[6] & Nicolas L Fawzi[1,2,*] (ID)

## Abstract

TDP-43 is an RNA-binding protein active in splicing that concentrates into membraneless ribonucleoprotein granules and forms aggregates in amyotrophic lateral sclerosis (ALS) and Alzheimer's disease. Although best known for its predominantly disordered C-terminal domain which mediates ALS inclusions, TDP-43 has a globular N-terminal domain (NTD). Here, we show that TDP-43 NTD assembles into head-to-tail linear chains and that phospho-mimetic substitution at S48 disrupts TDP-43 polymeric assembly, discourages liquid–liquid phase separation (LLPS) *in vitro*, fluidizes liquid–liquid phase separated nuclear TDP-43 reporter constructs in cells, and disrupts RNA splicing activity. Finally, we present the solution NMR structure of a head-to-tail NTD dimer comprised of two engineered variants that allow saturation of the native polymerization interface while disrupting higher-order polymerization. These data provide structural detail for the established mechanistic role of the well-folded TDP-43 NTD in splicing and link this function to LLPS. In addition, the fusion-tag solubilized, recombinant form of TDP-43 full-length protein developed here will enable future phase separation and *in vitro* biochemical assays on TDP-43 function and interactions that have been hampered in the past by TDP-43 aggregation.

**Keywords** amyotrophic lateral sclerosis; protein–protein interaction; RNA splicing; RNP granule; solution NMR spectroscopy
**Subject Categories** Neuroscience; Protein Biosynthesis & Quality Control; RNA Biology
**The EMBO Journal** (2018) 37: e97452

## Introduction

Amyotrophic lateral sclerosis (ALS) is a progressive neurodegenerative disease characterized pathologically by TDP-43 aggregates in a large majority of patient motor neurons (Arai *et al*, 2006; Neumann *et al*, 2006). As a canonical heterogeneous ribonucleoprotein (hnRNP), TDP-43 fills diverse roles in mRNA metabolism, localization, and transport (Buratti & Baralle, 2001; Kim *et al*, 2010; Ayala *et al*, 2011; Sephton *et al*, 2011; Tollervey *et al*, 2011; Alami *et al*, 2014; Ishiguro *et al*, 2016). To facilitate its many functions, TDP-43 has a multi-domain architecture consisting of a well-folded N-terminal domain (NTD) that mediates weak self-interaction (Chang *et al*, 2012; Mompean *et al*, 2016), two tandem RNA recognition motifs (Lukavsky *et al*, 2013), and an aggregation-prone predominantly disordered C-terminal domain (CTD) that mediates heterotypic interactions with binding partners (D'Ambrogio *et al*, 2009; Romano *et al*, 2014) using both glycine-rich low sequence complexity (LC) regions and helix–helix contacts to self-assemble via liquid–liquid phase separation (LLPS) (Conicella *et al*, 2016). This C-terminal region is cleaved and deposited in inclusions in ALS patient cells and in animal and cell culture models (Hasegawa *et al*, 2008; Budini *et al*, 2012; Clippinger *et al*, 2013), although the mechanistic link between CTD aggregation and cellular toxicity remains unclear (Johnson *et al*, 2009; Arnold *et al*, 2013).

Unlike the primarily disordered CTD, the TDP-43 NTD is highly conserved among animals (Appendix Fig S1; Ayala *et al*, 2005) and plays an essential role in TDP-43 splicing function (Zhang *et al*, 2013). Additionally, intact NTD is required for exogenous TDP-43 toxicity (Sasaguri *et al*, 2016) and for sequestration of endogenous TDP-43 in aggregates driven by expansion of an aggregation-prone region of the C-terminal domain (Romano *et al*, 2015). Together, these observations highlight the importance of NTD interactions

1   Department of Molecular Pharmacology, Physiology, and Biotechnology, Brown University, Providence, RI, USA
2   Graduate Program in Molecular Biology, Cell Biology and Biochemistry, Brown University, Providence, RI, USA
3   Department of Biochemistry, Stanford University School of Medicine, Stanford, CA, USA
4   Department of Structural Biology, St. Jude Children's Research Hospital, Memphis, TN, USA
5   Department of Pharmacology and Molecular Therapeutics, Uniformed Services University, Bethesda, MD, USA
6   Edward Doisy Department of Biochemistry and Molecular Biology, Saint Louis University School of Medicine, St. Louis, MO, USA
7   Neuroscience Graduate Program, Brown University, Providence, RI, USA
8   Department of Medicine, Stanford University School of Medicine, Stanford, CA, USA
*Corresponding author. Tel: +1 401 863 5232; E-mail: nicolas_fawzi@brown.edu
†These authors contributed equally to this work

with regard to cell function. However, little is known about the structure or contacts that mediate NTD self-interaction.

Mompean *et al* (2016) solved the solution NMR structure of monomeric TDP-43 NTD, revealing a tertiary structure reminiscent of the axin 1 DIX domain. Interestingly, DIX domains are capable of mediating self-assembly into dynamic, functional polymers via intermolecular head-to-tail interactions (Schwarz-Romond *et al*, 2007; Madrzak *et al*, 2015). We therefore hypothesized that the self-interaction previously reported for TDP-43 NTD might be a head-to-tail polymerization. Here, we sought to characterize NTD self-assembly using a multifaceted biophysical and cell biological approach.

We explore the self-assembly of globular TDP-43 NTD into high-order polymers and the effect of phosphomimetic substitution at a known post-translational modification site which we sought to validate by creation of a phospho-specific pS48 TDP-43 antibody. Using a novel construct and purification approach to generate soluble recombinant full-length TDP-43, we tested the effect of NTD polymerization on LLPS *in vitro*. With complementary cell-based assays, we measure the effect of the phosphomimetic substitution on in cell TDP-43 LLPS fluidity and splicing efficiency. Guided by chemical shift perturbation mapping and paramagnetic relaxation enhancements, we engineer two TDP-43 variants on distinct interfaces that when mixed are capable of saturating an NTD dimer state for which we solve the solution NMR structure.

## Results

### TDP-43 N-terminal domain self-assembly is disrupted by phosphomimetic variant S48E

Several groups have reported that TDP-43 NTD mediates assembly into dimers or other higher-order structures (Shiina *et al*, 2010; Chang *et al*, 2012; Wang *et al*, 2013), though the details of the assembly have only recently begun to be visualized with atomistic resolution (Afroz *et al*, 2017; Jiang *et al*, 2017; Mompean *et al*, 2017; Tsoi *et al*, 2017). Recombinant TDP-43 NTD (in our case: residues 1–80, with only three additional N-terminal residues, GHM, resulting from the cloning strategy and cleavage with TEV protease) is highly soluble when expressed and purified from bacteria. However, as described previously, increasing concentrations of NTD result in size-exclusion chromatography (SEC) profiles smoothly shifting to smaller elution volume, which is consistent with higher apparent molecular weight (Chang *et al*, 2012). This continuous shift of a single peak that becomes more asymmetric at higher concentrations is consistent with weak (μM) interactions. Correspondingly, concentrations above 20 μM resulted in altered NMR chemical shifts and a decrease in NMR signals. These concurrent changes in chemical shifts, intensities, and the resulting low signal-to-noise ratio precluded previous direct characterization of the assembled state by NMR (Chang *et al*, 2012).

Inspired by the observation that pentamerization of nucleophosmin is regulated by phosphorylation (Mitrea *et al*, 2014, 2016), we investigated proteomic databases to determine whether weak self-assembly of TDP-43 might be regulated by post-translational modification. TDP-43 NTD contains one reported serine/threonine phosphorylation, pS48, which was found in multiple phosphoproteomic analyses of various cell lines (Rigbolt *et al*, 2011; Hornbeck *et al*, 2012, 2015). This site is highly conserved as a serine or threonine in all organisms with TDP-43 orthologues examined except for *Xenopus* species (Appendix Fig S1A). In order to confirm that S48 is phosphorylated in cells, we created a custom polyclonal antibody serum against residues G40-G53 of TDP-43 that is specific to the peptide containing phosphorylated serine 48 (α-TDP-43 pSer48) (Fig 1A). HEK 293T cell lysate probed with α-TDP-43 pSer48 yields a band with a band at 43 kDa consistent with TDP-43 (Fig 1B). Furthermore, phosphatase treatment of cell lysate or fractions immunoprecipitated with commercial α-TDP-43 antibody abrogates α-TDP-43 pSer48 immunoreactivity (Fig 1C), confirming that our custom serum is specific for phosphorylated TDP-43 and suggesting that a small though measurable population of TDP-43 is constitutively phosphorylated in these cells, consistent with the previous phosphoproteomic results. Because this position is in a region that showed large NMR chemical shift differences upon increasing concentration of protein, consistent with NTD oligomerization at this site (Fig 1G), we tested the hypothesis that phosphorylation could alter assembly using a phosphomimetic substitution, S48E. Surprisingly, SEC profiles of TDP-43 NTD S48E are dramatically shifted from wild-type, showing a longer retention time and symmetric elution profile consistent with a monomer (Fig 1D) and nearly complete lack of self-assembly as indicated by a lack of concentration-dependent chemical shift changes (Fig 1H). S48E TDP-43 NTD appears well folded and nearly unchanged compared to the wild-type by fingerprint ($^1$H $^{15}$N HSQC) NMR spectra (see below), suggesting that the lack of assembly is not due to unfolding or adoption of a different global structure. Therefore, these data are consistent with the region near S48 playing a role in assembly.

### TDP-43 NTD assembles into linear chains with a monomeric repeat unit

The disappearance of NMR signals and the continued elution peak shift in SEC as a function of concentration observed previously (Chang *et al*, 2012) suggests that the assembly formed by TDP-43 NTD may be larger than a dimer. To determine the number of monomers making up the oligomeric form, samples were loaded onto SEC columns equipped with UV absorption and multi-angle light scattering (SEC-MALS) detectors at increasing concentrations. For TDP-43 NTD at 150 mM NaCl, the elution volume decreases continuously with increasing injection concentration (Appendix Fig S1D) as observed previously for TDP-43 NTD (Chang *et al*, 2012). The molar mass monitored over the eluting protein indicates an equilibrium between at least monomers and dimers. To test whether the NTD self-associates into larger species than dimers, we performed *in vitro* cross-linking experiments (Marzahn *et al*, 2016) of TDP-43 NTD WT samples with the primary amine cross-linker BS3. The cross-linked products were separated on SDS–PAGE and appear as ladders of increasing molecular weight, indicating the degree of oligomerization. Cross-linking was observed for all samples spanning concentrations of 10–200 μM. At low concentrations, monomers and dimers are observed while species up to heptamers are present at higher concentrations (Appendix Fig S1G). The NTD therefore self-associates into higher-order species and populates different oligomer sizes simultaneously.

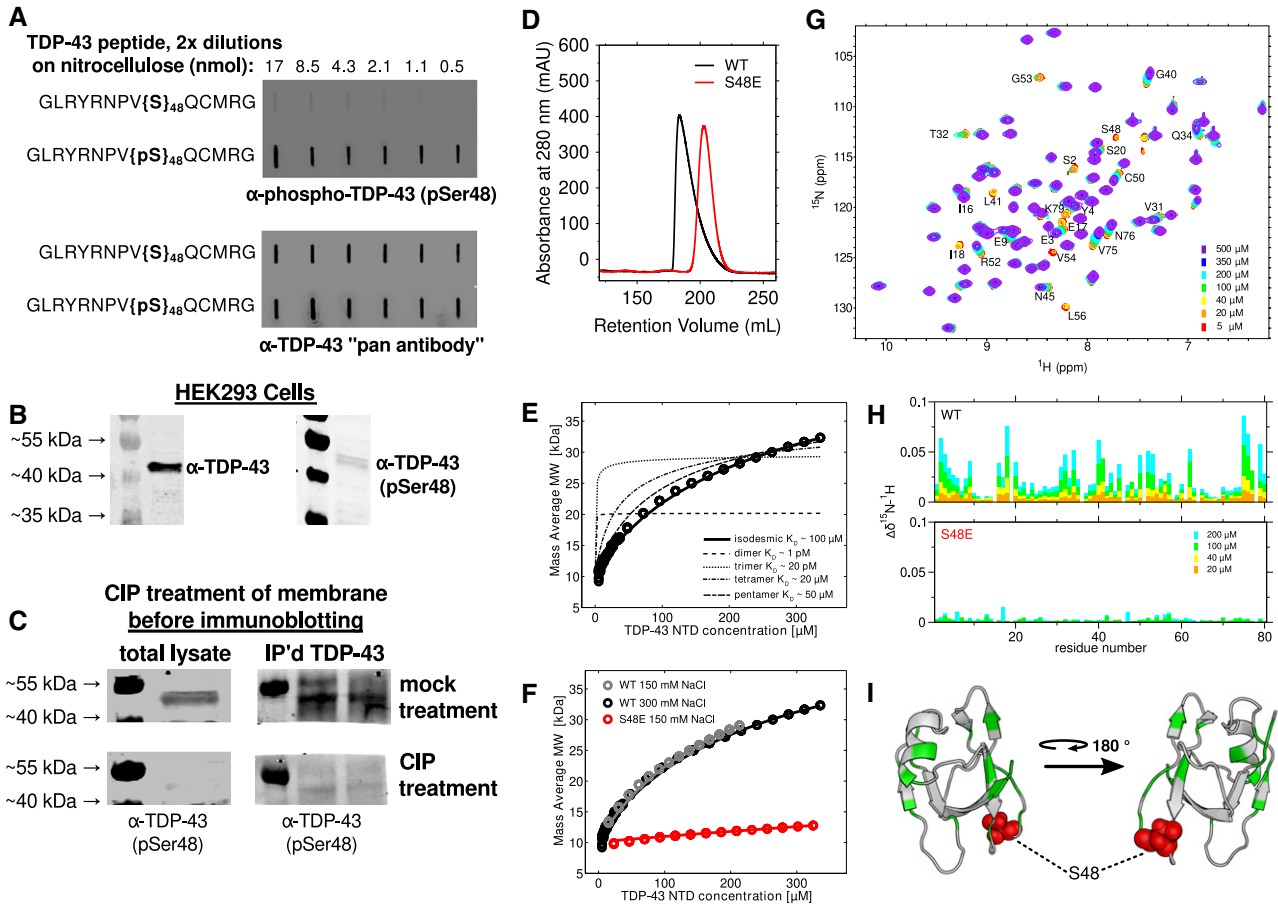

**Figure 1. TDP-43 NTD high-order oligomerization is impaired by S48E.**

A   Peptides composed of TDP43 (40–53), with and without phosphorylated Ser48, were serially diluted and spotted to nitrocellulose membranes. Polyclonal antibody (α-TDP-43 pSer48) specific to the phosphorylated peptide was used in the top panel showing specificity for pS48, and α-TDP-43 "pan antibody" recognizing the same peptide irrespective of phosphorylation was used in the bottom panel.

B   The α-TDP-43 pSer48 antibody and commercial TDP-43 antibody used in Western blots of HEK293T cell lysates both show reactivity at ~43 kDa, consistent with TDP-43 SDS–PAGE migration.

C   Standard Western blotting was performed on HEK293T cell lysates that had been transferred onto nitrocellulose membranes, except calf intestinal phosphatase (CIP, bottom) or a mock treatment (top) was used to treat the membranes prior to immunoprobing with α-TDP-43 (pSer48). Whole HEK293T cell lysates were used in the left panel. In the right panel, TDP-43 was first immunoprecipitated using commercial α-TDP-43 antibody prior to Western blotting.

D   Gel filtration chromatogram of 200 μM wild-type (black) and S48E (red) TDP-43 NTD. The shorter retention time and skewed profile of wild-type NTD is consistent with self-assembly. The single-point variant S48E results in a symmetric peak at longer retention time, consistent with predominantly monomer.

E   CG-MALS derived mass average as a function of increasing TDP-43 NTD concentration data are fit to an isodesmic self-association model (bold black line) with $K_D \sim 95$ μM. Fits for dimer, trimer, tetramer, and pentamer models are poor (dashed lines), shown for comparison.

F   CG-MALS data for wild-type are effectively the same at 150 mM (black, repeated from E for clarity) and 300 mM (gray) NaCl. S48E at 150 mM NaCl (red) shows dramatically disrupted assembly with $K_D \sim 2,000$ μM.

G, H   The concentration-dependent chemical shift deviations of $^1$H-$^{15}$N HSQC are large for wild-type and small for S48E TDP-43 NTD, consistent with disrupted binding. The CSDs are measured for 200 μM (cyan), 100 μM (green), 40 μM (yellow), and 20 μM (orange) WT compared to a monomeric control: 5 μM. For S48E, only 200 and 100 μM are shown.

I   The chemical shift deviations (at 100 μM with a cutoff of 0.02 ppm, shown in green) map to two different sides of TDP-43 NTD (PDB 2N4P), supporting a view that TDP-43 can assemble into linear chains via multiple interfaces. S48 is highlighted with red spheres.

To characterize the nature of this higher-order self-association of TDP-43 NTD, we used the quantitative equilibrium methods composition gradient multi-angle light scattering (CG-MALS) and sedimentation equilibrium analytical ultracentrifugation (AUC). The results from both methods support a model of weak infinite linear self-association, specifically isodesmic assembly resulting in unbranched linear chains. CG-MALS data were collected for wild-type TDP-43

NTD in a wide range of concentrations up to more than 300 μM. The mass average molecular weight increases continuously as a function of the concentration of the wild-type NTD (Fig 1E). Models for oligomerization into discrete dimers, trimers, or tetramers do not fit the data well. By contrast, an excellent fit can be achieved with an isodesmic self-association model, in which each addition of a monomer to pre-existing oligomers/polymers occurs with identical

affinity, here with values of 95 µM for wild-type NTD in the presence of 150 and 300 mM NaCl (Fig 1F). Higher-order assembly is similar at the two different ionic strength conditions, suggesting that electrostatic interactions are not primarily responsible for stabilizing the polymeric NTD states. These results suggest self-assembly of NTD monomers into linear polymers, with long polymers (up to 7–10mers at 300 µM protein concentration), short polymers, and monomers populated simultaneously in equilibrium (Appendix Fig S1B and C). S48E disrupts polymerization (Fig 1F, red), resulting in a $K_D$ of approximately 2 mM assuming an isodesmic assembly process.

We observe evidence of the same self-association behavior for TDP-43 NTD in AUC experiments. In sedimentation velocity (SV) experiments, at least three species are observed when the data are modeled to determine the sedimentation coefficient distribution of the present species, and their fractional populations shift to larger species in a concentration-dependent manner (Appendix Fig S1E). However, interconversion of multimeric species with different and unknown shape asymmetries precludes straightforward quantitative interpretation of SV-AUC fits. Sedimentation equilibrium (SE) AUC profiles, which do not suffer from assumptions of shape, for TDP-43 NTD WT protein in 150 mM NaCl fit well to an isodesmic self-association model with a $K_D \sim 40$ µM (Appendix Fig S1F). The discrepancy between the values calculated by CG-MALS and SE-AUC is likely due to the difficulty of obtaining the precise equilibrium constant for a weak association.

## NTD oligomerization contributes to TDP-43 full-length phase separation *in vitro*

We next sought to understand the role of TDP-43 NTD oligomerization in the context of the full-length protein. As described above, TDP-43 NTD is highly soluble even up to 20 mg/ml in our hands. Although it self-assembles into chains (Fig 1 and Appendix Fig S1), these interactions are relatively weak and NTD does not aggregate or undergo LLPS at any conditions tested. Conversely, TDP-43 C-terminal domain alone is sufficient to undergo LLPS at > 5 µM concentrations (Conicella *et al*, 2016). Therefore, we tested the hypothesis that TDP-43 NTD polymerization enhances phase separation of full-length TDP-43 by providing additional interaction sites creating cooperativity between NTD self-interactions and CTD self-interactions. Supporting this notion is the recent finding that a TDP-43 variant (where the RNA-binding RRM domains are replaced by GFP, leaving intact NTD and CTD) readily undergoes LLPS in cells (Schmidt & Rohatgi, 2016). However, difficulties in purifying soluble, monomeric full-length TDP-43 have so far prevented detailed biochemical characterization of TDP-43 full-length phase separation *in vitro* (Molliex *et al*, 2015). Our own attempts using the N-terminal maltose-binding protein (MBP) solubilizing fusion tag that worked for the related ALS-associated aggregation-prone protein FUS (Burke *et al*, 2015) were not successful. Therefore, we instead created a bacterial expression construct with a TEV cleavable C-terminal MBP fusion, placing the solubilizing MBP protein directly adjacent to the aggregation-prone C-terminal domain. We combined this refined strategy with the purification of the fusion protein under high salt conditions to reduce nucleic acid binding and further promote solubility. Purification of this construct via column nickel affinity chromatography followed by SEC results in a

significant fraction of the fusion protein eluting at a position consistent with monomer or low-order oligomer (Appendix Fig S2A). Immediately of interest, the low molecular weight fraction of full-length, wild-type TDP-43 elutes with a skewed peak consistent with assembly, while the same region of the full-length S48E fusion elutes with a symmetric peak at a longer retention time, consistent with monomeric NTD (Appendix Fig S2A). This observation directly mimics what we observed for TDP-43 NTD alone (Fig 1D), suggesting that S48E disrupts the dynamic equilibrium for full-length TDP-43 fusion protein assembly. Importantly, these fractions of both full-length wild-type and S48E are essentially devoid of nucleic acid as measured by ratio of UV absorbance at 260–280 nm, suggesting they will be useful for evaluating the behavior of full-length TDP-43 assembly and binding.

TDP-43 is a highly abundant protein, present at up to 3 µM in human cell lines assuming a uniform distribution across an entire cell volume (Ling *et al*, 2010; Beck *et al*, 2011), and known to be even more concentrated in the nucleus where it is primarily localized (Neumann *et al*, 2006). We therefore tested the ability of full-length purified low molecular weight/monomeric TDP-43 to undergo LLPS at low micromolar range concentrations. Addition of TEV protease to cleave the solubilizing C-terminal MBP fused to 2.5 µM full-length TDP-43 wild-type results in the formation of turbidity at physiological salt concentration (150 mM NaCl) (Fig 2A). Turbidity is absent before TEV addition (Appendix Fig S2B, time zero), and the increase in turbidity is due to robust phase separation, as observed by the formation of round liquid micron-sized phases seen by differential interference contrast (DIC) microscopy (Fig 2B). In contrast to wild-type full-length TDP-43, the single-point variant S48E fails to phase separate at these conditions (Fig 2A, top B), suggesting that TDP-43 NTD polymerization contributes to full-length TDP-43 phase separation. Phase separation of full-length TDP-43 S48E (and wild-type) can be observed at higher protein concentrations (Fig 2B, bottom), implying that the S48E variant increases the critical protein concentration required for phase separation rather than disrupt it entirely. These data support a model where the C-terminal domain, which can readily phase separate on its own, is primarily responsible for mediating a network of multivalent contacts and that additional interactions between NTDs enhance phase separation. Interestingly, as observed for the C-terminal domain alone (Conicella *et al*, 2016), after 2-h incubation these assemblies appear irregularly shaped, no longer flowing and fusing, consistent with aggregation within the LLPS state (Fig 2B).

## NTD oligomerization contributes to TDP-43 in cell phase separation and RNA splicing

Ribonucleoprotein granules and membraneless organelles assembled via LLPS of their components have been proposed to act as functional centers for RNA processing (Brangwynne *et al*, 2009, 2011; Hyman *et al*, 2014). Our data above suggest that polymerization of the TDP-43 NTD may contribute to this function/process as well. Indeed, it has been pointed out recently that the cooperation and interactions between the various domains of RNA-binding proteins have only begun to be systematically probed (Wei *et al*, 2016). We thus sought to test the effect of NTD polymerization on TDP-43 assembly and function in cells. Using a reporter assay for

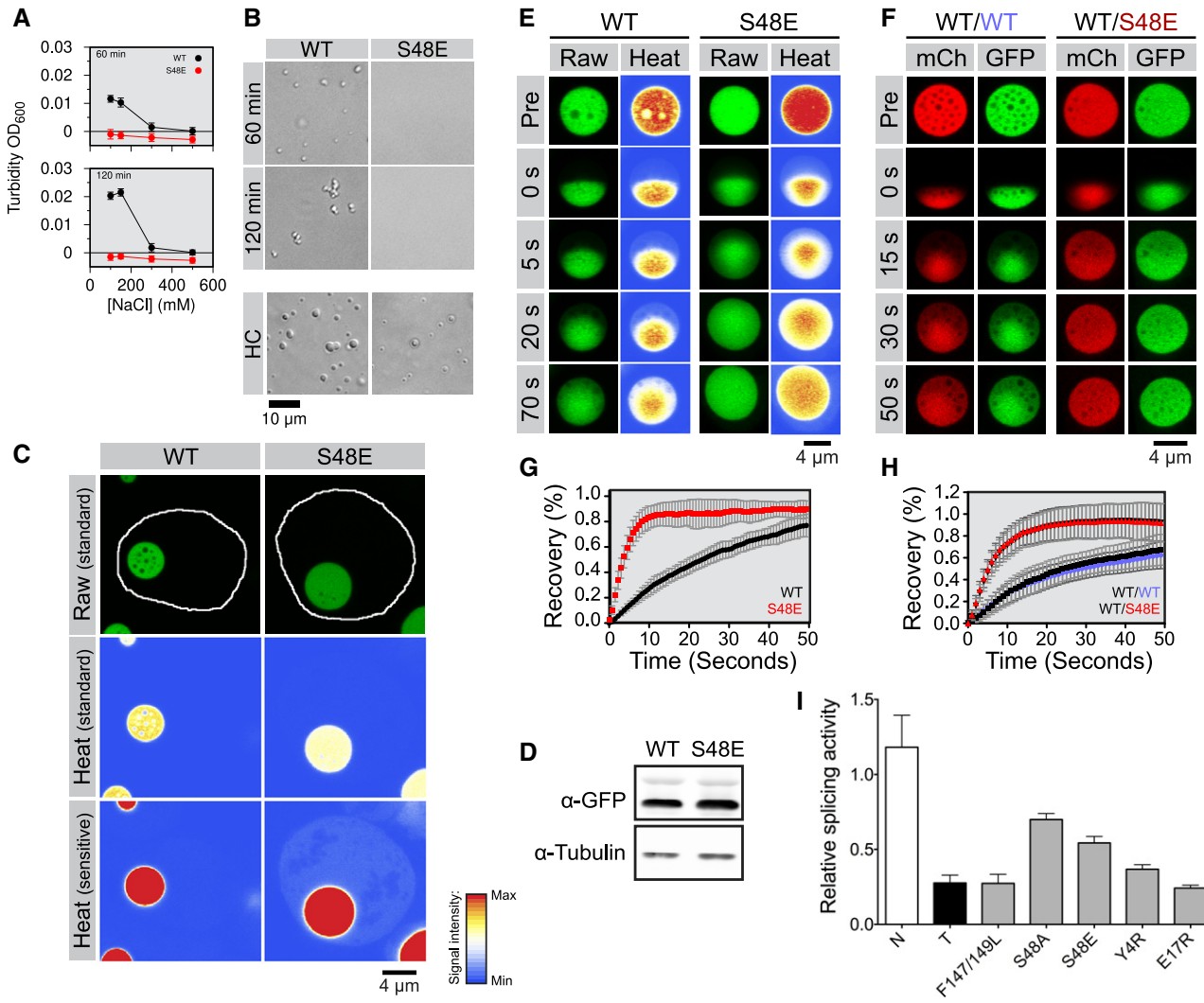

**Figure 2.   NTD polymerization ability contributes to TDP-43 phase separation and splicing.**

A    Turbidity of 2.5 μM wild-type (WT) and S48E TDP-43-MBP after 60 min (top) and 120 min (bottom) of incubation with TEV protease is consistent with phase separation at low salt concentration for the wild-type but phase separation is absent for S48E.

B    Differential interference contrast micrographs of 2.5 μM full-length TDP-43 MBP in 150 mM NaCl (top panel) after 60 and 120 min of incubation with TEV protease. WT shows phase separation, but S48E does not until the concentration is raised (HC).

C    Wild-type (WT) and variant (S48E) TDP-43$_{RRM-GFP}$ reporters form spherical, micron-sized nuclear droplets after overnight expression in 293T cells. Nuclei are outlined in white in representative raw images, and heat map representations of the signal intensities measured with standard and sensitive detector settings are provided below to highlight the differences in the nuclear TDP-43$_{RRM-GFP}$ reporter signal.

D    Immunoblot showing total expression levels of WT and S48E reporters in 293T cells. See Appendix Fig S2C for full immunoblots with molecular weight markers.

E    Representative time-dependent fluorescence recovery after half-droplet bleaching shows that S48E enhances intra-phase diffusion dynamics (decreases viscosity) of TDP-43 reporter particles.

F    The phosphomimetic S48E mutation has a dominant, fluidizing effect on the liquid dynamics of composite droplets, as revealed by half-bleach experiments of composite TDP-43 droplets formed by co-expression of wild-type TDP-43$_{RRM-mCherry}$ (red fluorescent, mCh) and the WT (left, blue type) or S48E (right, red type) TDP43$_{RRM-GFP}$ variants (green fluorescent, GFP) in 293T cells.

G, H    Quantification of fluorescence recovery after half-droplet bleaching of (G) GFP wild-type (black curve) and GFP S48E (red curve) or (H) mixtures of mCherry WT (black squares) plus GFP wild-type (blue circles) or mixtures of mCherry wild-type (inverted triangles) plus GFP S48E (red triangles). Error bars indicate s.d. of 20 measured particles from two biological replicates.

I    Relative splicing activity of CFTR exon 9 minigene reporter in control (N), TDP-43 siRNA knock-down (T) HeLa cells, RNA-binding-deficient mutant of F147/149L and TDP-43 NTD variants was calculated as the ratio of percent exon inclusion relative to WT. Levels of exon inclusion using the CFTR exon 9 minigene reporter were quantified as percent of exon inclusion from Appendix Fig S2G. RNAi-resistant wild-type TDP-43 (WT) and mutants were expressed in siRNA-treated cells. Error bars indicate s.d., $n \geq 4$.

TDP-43 intranuclear phase separation (Schmidt & Rohatgi, 2016), we determined the contribution of NTD interactions on TDP-43 phase formation by comparing wild-type and S48E forms of TDP-43 where the RNA-binding domains are replaced by GFP, termed here TDP-43$_{RRM-GFP}$. As described previously, when expressed in human (293T) cells, TDP-43$_{RRM-GFP}$ localizes to the nucleus and

concentrates into a single large liquid–liquid phase separated assembly per cell over time (Schmidt & Rohatgi, 2016; Fig 2C and D). We hypothesized that the phosphomimetic S48E variant would alter the assembly behavior and phase properties of these model TDP-43 phases. When expressed under identical conditions (Fig 2E), S48E TDP-43$_{RRM-GFP}$ also forms nuclear assemblies; however, these phases differ from wild-type TDP-43$_{RRM-GFP}$ in several critical parameters. Most strikingly, the S48E TDP-43$_{RRM-GFP}$ phase is much less viscous than those made by the wild-type TDP-43$_{RRM-GFP}$ (Fig 2E and G). This observation is consistent with reduced inter-protein contacts for S48E compared to wild-type TDP-43$_{RRM-GFP}$. We also find that this effect of S48E on droplet viscosity appears to be "dominant-negative"—when S48E and wild-type are co-expressed (Fig 2F and H), S48E/wild-type TDP-43 mixed phases show rapid kinetics like S48E-alone phases, suggesting that they are well mixed at the molecular level and co-constitute the phase. The dominant fluidizing effect of S48E suggests that wild-type NTD normally engages in chain-forming interactions that are disrupted in the presence of co-expressed S48E variant. Additionally, although qualitatively the phases formed by S48E and wild-type alone appear similar [and expression levels are indistinguishable (Fig 2D and Appendix Fig S2C and D)], quantitatively S48E TDP-43$_{RRM-GFP}$ partitions less readily into these phases and forms a slightly less dense concentration of TDP-43 inside the phase, as seen by the relative quantification of TDP-43$_{RRM-GFP}$ fluorescence intensity in the surrounding nucleoplasm and inside the phase, respectively (Appendix Fig S2E). Given that the wild-type NTD appears to stabilize TDP-43$_{RRM-GFP}$, we also tested the hypothesis that phase separation would occur earlier in wild-type TDP-43$_{RRM-GFP}$ transfected cells compared to S48E. Although transfection heterogeneity precluded quantification of kinetics, wild-type TDP-43$_{RRM-GFP}$ does form liquid phases earlier than S48E in a time course after transfection (see Appendix Fig S2F for representative images). Taken together, these data suggest that NTD polymerization contributes to TDP-43 phase separation in cells.

Given that the phase separation of TDP-43 in cells is altered by the S48E variant that controls TDP-43 NTD polymerization, we tested whether TDP-43 function is also altered by S48E. Previously, a correctly folded NTD has been shown to be important for TDP-43 splicing function (Zhang et al, 2013). To determine the effect of the Ser48 phosphomimetic variant (S48E) on TDP-43 function, we analyzed TDP-43 splicing regulatory activity in HeLa cells using the well-established cystic fibrosis transmembrane conductance regulator (CFTR) exon 9 minigene reporter (Buratti et al, 2001; Ayala et al, 2006). The levels of splicing are quantified as the percent of exon inclusion (Appendix Fig S2G), and the splicing activity is measured as the ratio of percent exon inclusion relative to WT (Fig 2I). As expected, siRNA-mediated knock-down of TDP-43 greatly decreased the splicing activity (Fig 2I and Appendix Fig S2G). The RNA-binding-deficient mutant F147/149L (Buratti & Baralle, 2001) used as control failed to rescue TDP-43 activity. Compared to wild-type, S48E showed approximately 40% reduction in regulatory function (Fig 2I), similar to other variants disrupting TDP-43 NTD assembly, as recently demonstrated (Afroz et al, 2017; Jiang et al, 2017; Mompean et al, 2017). The corresponding Ala substitution (S48A) also decreased TDP-43 splicing regulatory activity, by approximately 25%, consistent with decreased self-association in vitro (see below). These findings suggest that the

details of the NTD structure and interactions in the region surrounding S48, not just phosphorylation at S48, are critical for mechanisms that mediate TDP-43 control of splicing. To further determine the requirement for NTD oligomerization in TDP-43 splicing regulatory function, we studied NTD Y4R and E17R, two other self-assembly deficient variants (see below), in our cellular assays. We observed a dramatic decrease in TDP-43 activity upon introduction of both mutations (Fig 2I) and in particular E17R, which shows approximately 80% loss in activity. These findings, combined with our structural data showing that S48A and S48E disrupt polymerization, strongly suggest that TDP-43 acts as a polymer or oligomer during splicing regulation and that this activity may be modulated by modification of S48 or other interfacial residues.

## Structural details of TDP-43 N-terminal domain head-to-tail polymerization

Because the polymerization of TDP-43 NTD regulates in cell phase separation and splicing function, we next sought to characterize the structural details of NTD polymerization and the structural mechanism by which interface modifications control assembly and function. Because the formation of a dynamic, high molecular weight NTD polymer makes direct NMR structural determination difficult (Chang et al, 2012), we began by first distinguishing the repeat unit for NTD polymerization in solution. The CG-MALS and NMR data we collected (Fig 1E and H) are consistent with a repeat unit of a monomer with two distinct interfaces. Therefore, one possibility is that TDP-43 NTD self-assembles into a linear chain with a head-to-tail assembly. Unlike a symmetric dimer where the same interface mediates self-interaction on both monomers, a head-to-tail assembly will have two interfaces—one interface, the "head", that forms contacts with the other monomer via a distinct region or second interface, "the tail". Above we demonstrated that S48E disrupts polymerization and lies in one interface identified by NMR chemical shift deviations of wild-type NTD (Fig 1H and I). Therefore, we expected that TDP-43 NTD S48E would only be able to bind the TDP-43 NTD wild-type interface that includes S48 and not the other interface. Hence, NMR spectra observing ($^{15}$N-labeled) TDP-43 NTD wild-type titrated with increasing concentrations of NMR-invisible (natural isotopic abundance) S48E TDP-43 NTD would show chemical shift deviations only near S48. Indeed, chemical shift deviations are only observed in the C-terminal half (residues 40–80) of wild-type (Fig 3A, top). The complementary experiment where increasing quantities of NMR-invisible wild-type combined with NMR visible ($^{15}$N-labeled) TDP-43 NTD S48E shows chemical shift deviations at residues in the N-terminal half (residues 1–35) of the S48E protein (Fig 3A, bottom). Therefore, this experiment identifies the residues in the N-terminal region as participating together in the second interface. Only residues 75–80 show significant chemical shifts in both titrations, suggesting these residues are near or connected to both interfaces. This paradox may be due to these residues forming part of the final β-strand that spans the width of the protein and contacts both interfaces. Hence, an allosteric connection may link residues 75–80 to both interfaces. Furthermore, the structure of NTD residues 1–77 solved by Mompean and coworkers (Mompean et al, 2016) (PBD ID: 2N4P) depicts V75 forming a hydrogen bond with residue of E9, the latter of which is in the

    

N-terminal interface. Therefore, the CSD observed in the titration of either $^{15}$N S48E with wild-type (or $^{15}$N S48E with Y4R, see below) may be the consequence of the conformational change caused by the interaction occurring in the N-terminal interface. These residues are the only outliers from the single interface behavior observed for other residues in the domain. Collectively, these data immediately point to a head-to-tail assembly (Fig 3B), where S48E reduces the affinity at the interface.

In order to further define the interface of the self assembled NTD, we also measured intermolecular $^1H_N$ paramagnetic NMR relaxation enhancements (PREs) that are high for positions in close proximity to a stabilized nitroxide spin label, even in transiently populated states. Assuming a two-state monomer–dimer equilibrium, the relative orientation of two NTD monomers in the assembled dimer state can be discerned given a sufficient number of samples with conjugated nitroxide radical spin labels at single positions spatially separated across the monomer surface. Therefore, MTSL spin labels were conjugated to the NTD at engineered cysteine variants at positions 2 or 29 or at native cysteine positions 39 or 50 and mixed with $^{15}$N isotopically enriched NTD C39S/C50S in order to measure $^1H_N$ transverse paramagnetic

relaxation enhancements, $\Gamma_2$, arising only from intermolecular contacts (Fig 3C). Addition of MTSL spin labels did not result in perturbations to the native monomeric structure as confirmed by $^1$H-$^{15}$N HSQC spectra (Appendix Fig S3A). Comparing the CSDs of each PRE variant to the diamagnetic spin label analog (AcMTSL)-labeled species suggested that cysteine variants and addition of spin labels did not disrupt the self-assembly for S2C and C50 and decreased affinity but maintained a similar pattern of chemical shifts compared to the reference for the variants of S29C and C39 (Appendix Fig S3B and C). The PRE measurements then provide complementary contact details, with C39 and C50 reporting on the C-terminal interface, and S2C and S29C reporting on the N-terminal interface. Interestingly, the cysteine-free C39S/C50S variant shows similar chemical shift patterns but approximately 3× weaker assembly compared to wild-type in reducing conditions (Appendix Fig S3B), suggesting that cysteine side chains have some contribution to assembly, though no evidence of disulfide dimers in the wild-type is observed, as expected for reducing (1 mM DTT) conditions.

For each MTSL spin label, we observed a specific and distinct pattern of PREs that could each be mapped to a distinct cluster of

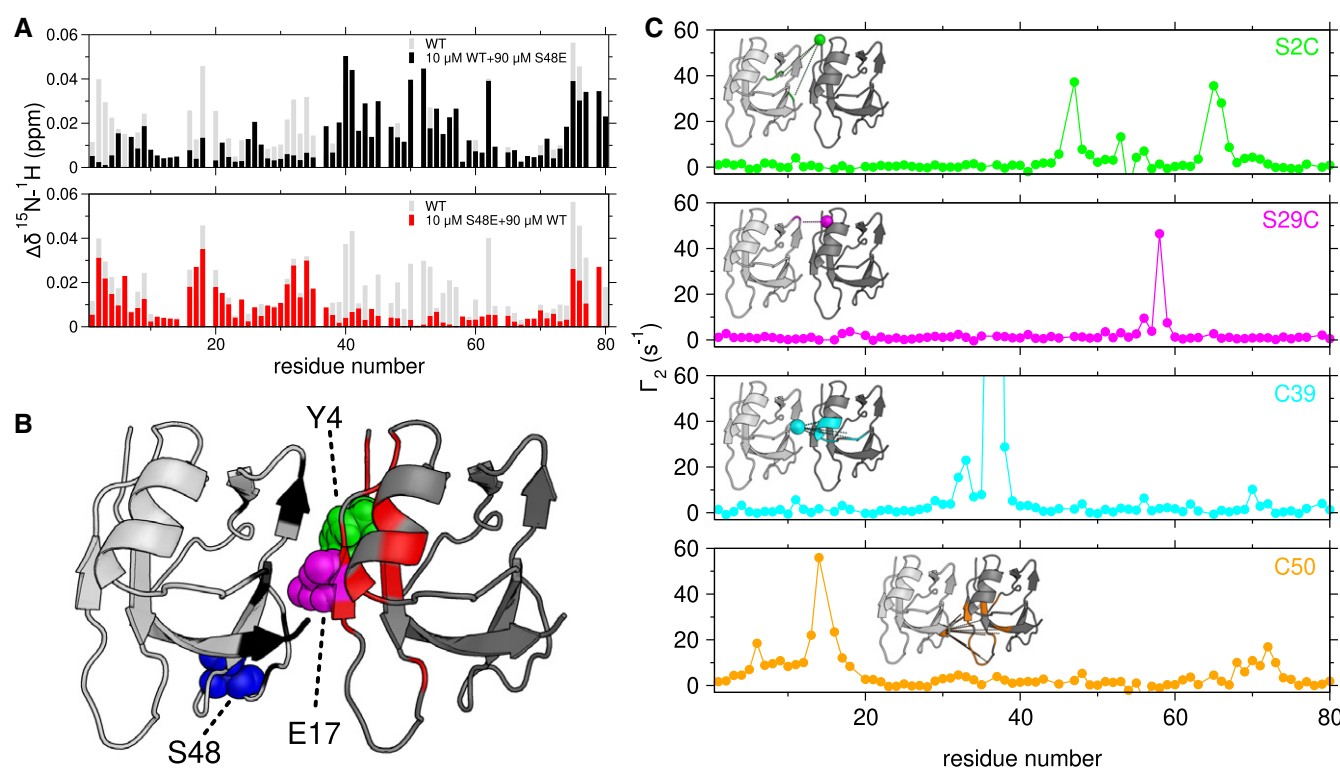

**Figure 3. Defining the TDP-43 NTD dimer interface.**

A   Chemical shift deviations compared to 10 μM alone measured for a mixture of 10 μM $^{15}$N wild-type (WT) and 90 μM S48E (upper, black) identifies the C-terminal interface, while 10 μM $^{15}$N S48E and 90 μM WT (lower, red) TDP-43 NTD identifies the N-terminal interface. The gray bars represent the CSDs of 100 μM wild-type alone compared to 5 μM wild-type (repeated from Fig 1E).

B   CSDs with cutoffs of 0.02 ppm (black) and 0.01 ppm (red) mapped on a head-to-tail dimer model created from NTD monomer structure (2N4P). The interface residues Y4 (green), E17 (magenta) and S48 (blue) are depicted as spheres.

C   $^1H_N$ PREs arising from a mixture of TDP-43 NTD C39S/C50S and TDP-43 NTD with a MTSL spin label at position S2C (green), S29C (magenta), C39 (cyan), or C50 (orange) provide structural constraints for a low-resolution dimer model (Inset). PREs with cutoff of 10 s$^{-1}$ mapped to the structure of the TDP-43 NTD.

residues on the NTD surface (Fig 3C). Data from C39 and S29C labels together suggest that the α-helix in N-terminal interface contacts with the β-strands in the C-terminal side, which forms the upper part of the interface. The PREs from C50 and S2C reveal that the lower part of the interface mainly involves the loop of N45-G53 from the C-terminal side and the antiparallel beta-strands on the N-terminal side. Using rigid-body docking (Clore & Schwieters, 2003; Zwolak *et al*, 2010) of the published solution NMR structure of residues 1–77 of the NTD monomer (PDB 2N4P) based on ambiguous distance restraints generated from CSP and PRE data, we generated a low-resolution model of the NTD dimer interface. The model that best fits the experimental data depicts a head-to-tail interface with approximately a translation of one monomer width (Fig 3B). The head interface is comprised of N-terminal residues including those in β-strands 1 and 2 as well as the α-helix. The tail interface, on the other hand, is comprised of C-terminal residues, including the β-strands and the S48-V54 loop.

### Structure of a stable TDP-43 NTD dimer by mixing two distinct interface-disrupting variants

Based on our model of the NTD dimer interface, we hypothesized that mutations introduced on the head interface (residues 1–30) would inhibit self-assembly as potently as S48E. In order to test this hypothesis, we generated NTD variants E17R and Y4R and examined their effects on self-assembly. When the NMR chemical shift deviations for increasing concentrations of each mutant alone were compared to wild-type, we found that E17R and Y4R NTD variants do not disrupt folding (Appendix Fig S4A) but do disrupt self-assembly as measured by SEC elution volume (Appendix Fig S4B), lack of NMR chemical shift deviations upon increasing concentration (Appendix Fig S4C), and disruption of splicing function (Fig 2I), consistent with E17 and Y4 positions being in the interface (Fig 3A and B). Given that Y4R and S48E each block distinct interfaces, we hypothesized that mixing these two variants together would result in saturation of the dimer state without the formation of higher-order oligomers. Furthermore, this dimer has an expected molecular weight of 20 kDa and therefore should be readily visible by standard solution NMR techniques. Titration of increasing concentrations of unlabeled Y4R into 100 μM NMR visible ($^{15}$N labeled) S48E resulted first in disappearance of peaks due to the kinetics of exchange between monomer and complex (Fig 4A), as observed in wild-type (Fig 1G). At millimolar concentrations of the binding partner, the resonances reappear with very large chemical shift differences from the unbound positions, consistent with saturation of a complex with micromolar affinity [as observed for EIN:HPr complex (Fawzi *et al*, 2010)]. Significant NMR chemical shift deviations occur on the N-terminal side of $^{15}$N S48E due to addition of Y4R and on the C-terminal side of $^{15}$N Y4R due to addition of S48E (Appendix Fig S4D). The residues with large CSD in the titration of $^{15}$N S48E with Y4R, such as S2 and T32 (Fig 4B left), did not experience significant shifts in the titration of $^{15}$N Y4R with S48E (Fig 4B right), which is additional evidence that there are two distinct interfaces involved in the formation of the asymmetric dimer. Together, these data demonstrate that S48E and Y4R form an asymmetric dimer complex, rather than polymers, that preserves the wild-type head-to-tail interface.

In order to determine the atomic resolution structure of the S48E/Y4R dimer complex, we performed $^{13}$C/$^{15}$N-edited and $^{13}$C/$^{15}$N-filtered/edited NOESY measurements using the $^{13}$C/$^{15}$N-labeled S48E (or Y4R) complexed with natural abundance Y4R (or S48E) (Appendix Fig S5A). The filtered NOESY allowed us to specifically probe the intermolecular contacts involved in the dimer interface. Most of intermolecular NOEs arise from the interfacial residues, S29, T30, A33, and Q34 on the α-helix and I16-P19 on the β-strand 2 in the N-terminal half and C39-L41, C50-G53 on β-strand 5, V54- L56 on β-strand 6 and terminus residues N76-D80 in the C-terminal half, consistent with the interface determined by CSDs and PREs (Fig 3B and Appendix Fig S5B). With the intra- and intermolecular NOEs used as distance restraints, we calculated the dimer structure constructed by Y4R and S48E, shown in Fig 4C. The α-helix (on the N-terminal half of S48E) lies against the β-strands on the C-terminal of Y4R. The first half of the α-helix, including residues S29 and T30, mainly contacts V54, R55, and L56 on β-strand 6 (Fig 4D). A33, on the second half of α-helix, contacts C39, G40L41 (Fig 4D). The residue at the end of α-helix, Q34, as well as adjacent F35 and P36 interacts with the C-terminal residues N76, P78, and K79 (Fig 4E). In the lower part of interface, the β-strand 2 of I16-P19 (N-terminal) forms a parallel sheet with β-strand 5, residues C50-G53 (C-terminal) (Fig 4F).

The structure of each subunit is largely consistent with the previously published NTD monomer structure of 2N4P (residues 1–77 Cα RMS deviation = 1.48 Å; Appendix Fig S5E; Mompean *et al*, 2016), except for deviations at β-strands 4 and 5 and the extreme C-terminal residues. Additionally, our NTD dimer structure is even more similar to a recent 2.1 Å resolution X-ray crystal structure of a TDP-43 NTD (5MDI) (Afroz *et al*, 2017) where the asymmetric unit comprises two nearly identical monomers (Cα RMS deviation of 0.6 Å) with a similar head-to-tail orientation which forms a polymer by crystallographic symmetry. The Cα RMS deviation of the individual Y4R and S48E subunits in our structure and the corresponding monomer subunits in the crystal structure is low, 0.899 and 1.357 Å, respectively, indicating agreement of the NTD monomer component structures. However, the overall Cα RMS deviation for our dimer and the asymmetric unit that comprises the dimer in the crystal structure is 2.033 Å. The larger deviation is due mainly to differences in the orientation of the interface (Appendix Fig S5D)—primarily a rotation of −17° about an axis approximately perpendicular to the surface and translation of ~2 Å (Appendix Fig S5D).

## Discussion

Assembly via low-complexity, disordered domain (Kato *et al*, 2012; Burke *et al*, 2015; Lin *et al*, 2015; Molliex *et al*, 2015; Nott *et al*, 2015; Patel *et al*, 2015; Conicella *et al*, 2016; Feric *et al*, 2016; Mitrea *et al*, 2016) and structured domain (Li *et al*, 2012; Conicella *et al*, 2016; Marzahn *et al*, 2016; Pierce *et al*, 2016) interactions has been proposed as a functional organizing principle of RNP granule assembly. Previous studies of TDP-43 NTD assembly suggested that the NTD is capable of self-assembly *in vitro* (Chang *et al*, 2012) and postulated that the fundamental, functional state of TDP-43 is a homodimer mediated by intermolecular interactions in the NTD (Wang *et al*, 2013; Zhang *et al*, 2013). The

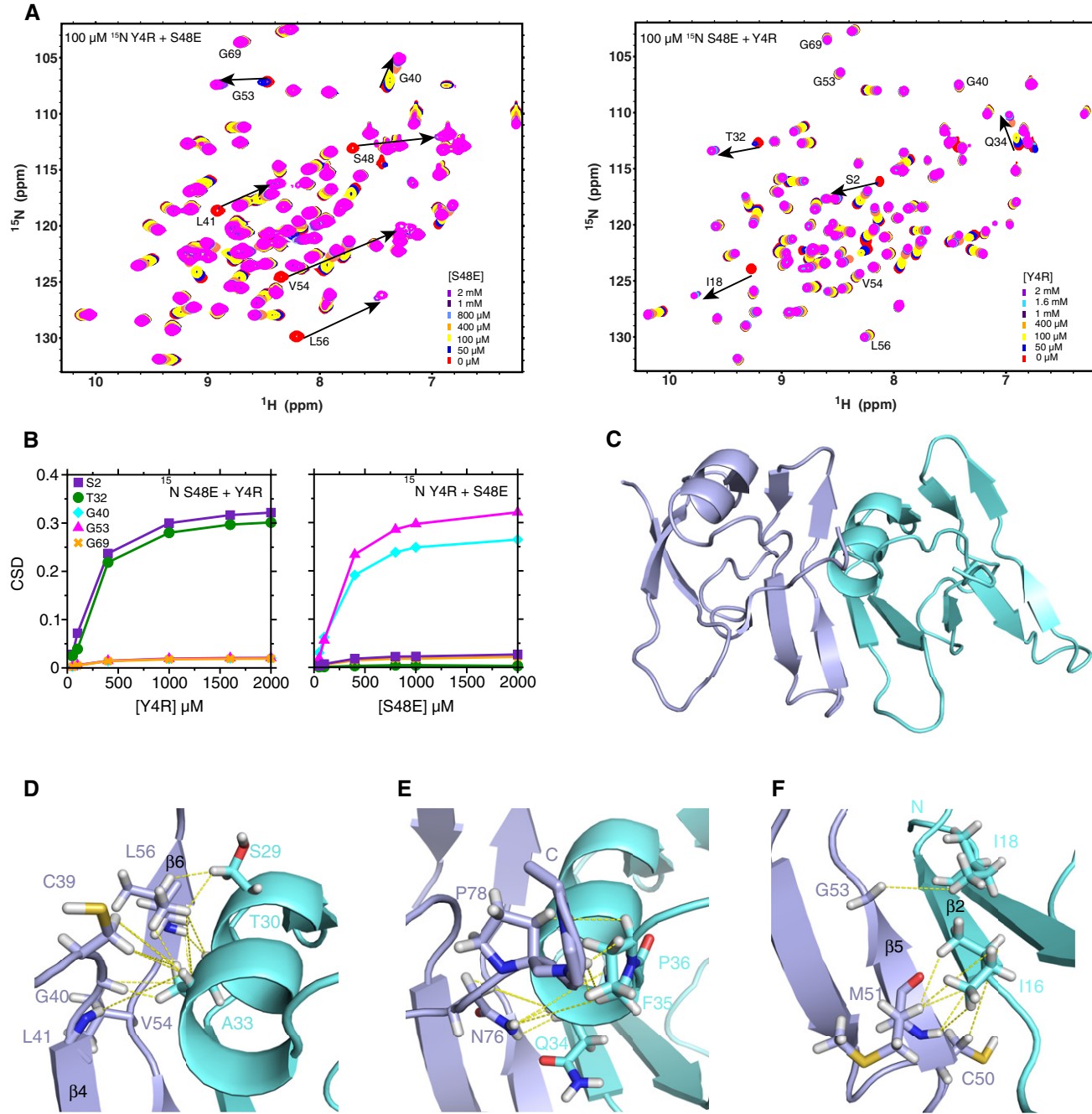

**Figure 4. A solution structure of dimeric TDP-43 NTD head-to-tail assembly.**

A    NMR HSQC spectra of 100 μM $^{15}$N-labeled S48E as a function of increasing unlabeled Y4R (right) and 100 μM $^{15}$N-labeled Y4R as a function of increasing unlabeled S48E (left) show resonances in the interface disappear due to exchange between monomeric and complex states, as observed in wild-type. However, peaks reappear after saturation of the complex (arrows), enabling observation of very large saturated chemical shift deviations consistent with a distinct chemical environment at the interface formed upon dimerization.

B    Chemical shift mapping of Y4R and S48E TDP-43 NTD illustrates the saturation of selected resonances of TDP-43 NTD dimer: S2 (purple square) and T32 (green circle) in one interface, G40 (cyan diamond) and G53 (magenta triangle) in the other, and G69 (orange x) in neither interface.

C–F  The dimer structure (C) solved for the complex of Y4R (blue) and S48E (cyan) (PDB 6B1G) where panels (D–F) represent the zoom-in views of the interface. Images of the representative regions highlight intermolecular NOEs shown as yellow dashed lines. The complete NOEs are shown in Appendix Fig S5C.

assembly of TDP-43 NTD into dimers, oligomers, or polymers has been the subject of several recent, conflicting reports. Our data supporting a head-to-tail chain are inconsistent with the models

that suggest symmetric dimerization (Jiang *et al*, 2017; Mompean *et al*, 2017), where the same interface mediates the interaction in both protomers in the dimer (related by C2 symmetry). However,

these studies were based primarily on identification of interfaces regions by mutagenesis disrupting splicing function and biophysical interactions. Given that these studies each identified a different region, L27-L28 and L71-V74, each of which is in or close to our N-terminal or C-terminal interface, respectively, we suggest that each study found one of two head-to-tail interfaces. In addition to mutagenesis, Mompean *et al* performed filtered NOESY NMR experiments similar to the one we used to support their symmetric dimer model. Although it is possible that different sample conditions (i.e., low pH in Mompean *et al*) alter the interface dramatically, it is also likely that dimer (and higher-order structures) is not saturated in their samples of polymerization competent wild-type, as evidenced by the smaller chemical shift differences compared to our NMR titration, complicating interpretation of filtered NOESY experiments which can contain spectral artifacts from incomplete filtering (Breeze, 2000; Gobl *et al*, 2014). Conversely, our structural finding of a head-to-tail assembly is largely in agreement with the interpretation of the crystal structure of a head-to-tail polymer of TDP-43 (Afroz *et al*, 2017). The interfaces identified are identical except for a slight rotation/translation. This difference in interface may arise either from deformations due to the S48E and Y4R variants we used, incomplete intermolecular restraints in our NMR structure, or alternatively from deformations due to crystallization. The crystals of TDP-43 NTD pack monomers into three intertwined superhelical filaments within a crystal lattice — hence, the polymer may be slightly deformed in order to fit an integral number of monomers (12) into a single helical turn necessary for accommodation into the Cartesian crystal lattice, in addition to potential changes due to crystal contacts between individual filaments. However, the importance of the details of the superhelical polymer is unknown as the functionally relevant range of polymer lengths populated in cells remains to be determined. The relatively weak NTD isodesmic $K_D$ we measure here, in general agreement with other NMR and biophysical studies (Jiang *et al*, 2017; Mompean *et al*, 2017), suggests that even at the high concentrations estimated for phase separated assemblies of TDP-43 (1 mM ~ 50 mg/ml) (Schmidt & Rohatgi, 2016), the great majority of the NTD would be expected to be incorporated in oligomers < 10mer (Appendix Fig S1B). This size distribution favoring low-order polymers is consistent with dynamic rearrangements expected for liquid-like behavior that is observed for TDP-43 assemblies by fluorescence recovery after photobleaching (FRAP) (Schmidt & Rohatgi, 2016). Conversely, long, stable polymers extending to the micron size scale in a dense droplet would not be expected to diffuse micron distances in seconds timescales in order to show rapid FRAP (Janke *et al*, 2017).

We previously showed that the predominantly disordered, aggregation-prone TDP-43 C-terminal domain readily undergoes LLPS via cooperation between intermolecular α-helix/α-helix and low-complexity glycine-rich region interactions (Conicella *et al*, 2016). Here, we demonstrate that full-length TDP-43 also self-assembles into LLPS droplets at physiological concentrations (Ling *et al*, 2010) and solution conditions *in vitro* via the multivalent C-terminal domain and enhanced by linear polymerization/oligomerization of the globular NTD. Importantly, we find that TDP-43 LLPS is tuned both *in vitro* and in a cell-based assay by the same single amino acid substitution mimicking post-translational modification that disrupts TDP-43 NTD polymerization (Fig 1).

Previous studies evaluating the role of ALS-associated mutations on TDP-43 have suggested that these variants decrease the fluidity of nuclear (Schmidt & Rohatgi, 2016) and cytoplasmic (Alami *et al*, 2014; Gopal *et al*, 2017) TDP-43 granules. Interestingly, the phosphomimetic substitution studied here decreases phase separation propensity *in vitro* and increases fluidity of LLPS TDP-43 in cells. Consistent with the idea that NTD-tuned TDP-43 assembly may be physiological regulated, our cellular splicing assays

## TDP-43 oligomers:
Functional high-order oligomers promote phase separation

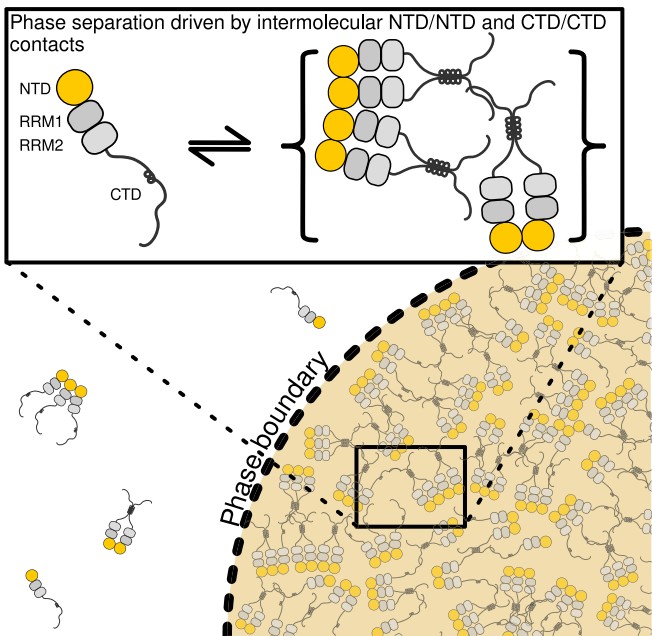

## NTD oligomerization-disrupting mutants:
Disruption of NTD/NTD contacts destabilizes phase separation

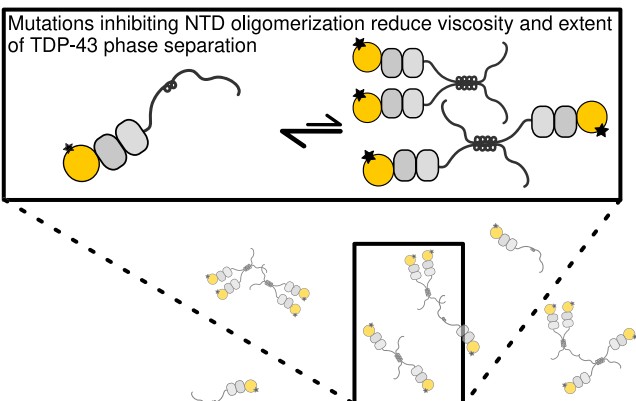

**Figure 5. A model for TDP-43 functional LLPS based on NTD and CTD contacts.**

Top: Intermolecular contacts between TDP-43 molecules via their NTD domains enhance phase separation of the CTD and partitioning into membraneless organelles by contributing to a network of interactions. Disruption of NTD oligomerization by mutagenesis of residues on the interaction interface (bottom) reduces the TDP-43 intermolecular interactions, subsequently reducing the extent of LLPS.

indicate that the same phosphomimetic variant S48E significantly impairs TDP-43 activity. Our observations that TDP-43 self-assembly is important for splicing regulation are in agreement with previous work suggesting the domain must be properly folded for TDP-43 function (Zhang *et al*, 2013). A previous study demonstrated that TDP-43 splicing activity is also inhibited by deletion of the helical region of the C-terminal domain (D'Ambrogio *et al*, 2009) which contributes to TDP-43 CTD phase separation (Conicella *et al*, 2016). Therefore, at least in the case of the well-established exon 9-CFTR system, TDP-43 inhibition of exon splicing requires both intact NTD and CTD contacts which we show each have unique contributions to phase separation.

An important caveat regarding the potential role of phosphorylation at S48 in directing function is worth noting. We establish here (i) that the NTD oligomerization is important for function, (ii) that phosphosite S48 (Rigbolt *et al*, 2011; Hornbeck *et al*, 2015) is conserved and phosphorylated at low levels in cells, and (iii) that a phosphomimic alters phase separation and function. In the future, it will be important to determine whether, and by what kinases, S48 is physiologically phosphorylated as a switch for TDP-43 function. Nevertheless, the functional contribution of TDP-43 NTD polymerization with regard to splicing raises the question of how cells can spatially and temporally regulate of TDP-43 function. Based on the similar monomeric structures and head-to-tail self-assembly of TDP-43 NTD to DIX domains, it is reasonable to hypothesize that similar mechanisms may regulate their assembly. Indeed, the Disheveled 2 DIX domain polymerization is inhibited post-translationally by the addition of ubiquitin to multiple lysine residues (Madrzak *et al*, 2015), and both the axin 1 and Disheveled 2 DIX domains can be post-translationally phosphorylated (Trinidad *et al*, 2012; Yi *et al*, 2014; Mertins *et al*, 2016). Furthermore, phosphorylation of other domains of TDP-43 has been shown to tune function (Li *et al*, 2017). Therefore, future efforts examining the biophysical and cellular effects of TDP-43 NTD phosphorylation will be useful to determine the pathways regulating TDP-43 self-assembly and function.

Our data are consistent with a model of TDP-43 interactions where the NTD binds other NTD in a chain, working together with the multivalent CTD to bring together monomers of TDP-43 into phase separated complexes (Fig 5, model). We do not find evidence here that the long, highly dynamic C-terminal domain (Conicella *et al*, 2016) connected by a dynamic linker to the NTD and RRMs via a dynamic linker (residues 80–100; Qin *et al*, 2014; Mompean *et al*, 2017) could be kept distant from each other upon NTD oligomerization, as recently proposed (Afroz *et al*, 2017). Although these authors showed that cells transfected with TDP-43-GFP fusions are more likely to form GFP-positive inclusions when the TDP-43 NTD interface is disrupted, the mechanism increasing aggregation is unknown. Further work in cells and organisms using endogenous TDP-43 will be important to determine whether variants or post-translational modifications that disrupt TDP-43 self-interaction can also alter TDP-43 aggregation and its associated toxicity in ALS.

Finally, we anticipate that the nucleic acid-free, soluble recombinant full-length TDP-43 fusion protein and the TDP-43 phase separation assays described here will be of use in the future for evaluating the mechanistic effect of ALS variants, RNA interaction, and intermolecular interactions that give rise to TDP-43 function.

## Materials and Methods

### Cloning, expression, purification of recombinant proteins

The DNA sequence encoding TDP-43 with an N-terminal Thioredoxin leader sequence, 6xHis (Thio-His) tag, and TEV cleavage site (Peti & Page, 2007) was codon optimized and synthesized by DNA2.0 in the pJ411 bacterial expression vector (designated pJ411/TDP-43). In order to generate the expression construct for the TDP-43 NTD, the codon for residue N81 was mutagenized by Quikchange (Agilent Genomics) to a stop codon.

The expression vector containing a cleavable C-terminal maltose-binding protein (MBP) fusion tag (designated pJ4M) was constructed using overlap extension PCR (Horton *et al*, 1989) of MBP and pJ411 DNA fragments. The DNA sequence encoding for MBP was amplified from an N-terminal MBP fusion construct optimized for bacterial expression (Peti & Page, 2007) using primers 5-TTCCAGGGTAGCGGTACCAAAATCGAAG-3 and 5-GGGTTATGCT AGGGGGTCAATGATGGTGATGGTGATGGGTACCGCCTTCAGC-3. The pJ411 DNA sequence was amplified from the pJ411/TDP-43 construct using primers 5-GCTGAAGGCGGTACCCATCACCATCACC ATCATTGACCCCCCTAGCATAACCC-3 and 5-GTACAGGTTCTCCTCG AGTCATCACATATGGGTATATCTCCTTCAAAAGTTAAAC-3.

The TDP-43-MBP expression construct (designated pJ4M/TDP-43) was created by subcloning the DNA sequence for TDP-43, which was amplified from pJ411/TDP-43 by PCR using primers 5-TA ATACGACTCACTATAGGG-3 and 5-TCATCACTCGAGCATGCCCC-3, into pJ4M via restriction sites NdeI and XhoI.

Point mutations were introduced either by Quikchange mutagenesis or by overlap extension PCR with mutagenic primers. Proteins were overexpressed in *Escherichia coli* BL21 Star DE3 (Life Technologies). Cells were grown in LB media for natural abundance isotope labeling. For expression of [U-99% $^{13}$C, 99% $^{15}$N] isotope labeling, cells were grown in M9 minimal media supplemented with $^{13}$C-glucose and/or $^{15}$NH$_4$Cl (Cambridge Isotope Laboratories). In order to purify the TDP-43 NTD, cell cultures were grown at 37°C to an OD600 of 0.6–0.9. Protein expression was induced by addition of isopropyl β-D-1-thiogalactopyranoside (IPTG) (GoldBio) to 1 mM, followed by incubation at 37°C for 4 h. Cells were harvested by centrifugation, resuspended in NTD binding buffer (20 mM Tris–Cl pH 8.0, 500 mM NaCl, 10 mM imidazole, 1 mM DTT) supplemented with complete EDTA-free protease inhibitor cocktail (Roche), and lysed using an EmulsiFlex C3 (Avestin). Cell lysates were clarified by centrifugation and applied to a 5 ml Histrap HP column (GE Healthcare). The column was washed with five volumes of binding buffer, and TDP-43 NTD was eluted with a linear gradient of NTD elution buffer (20 mM Tris–Cl pH 8.0, 500 mM NaCl, 500 mM imidazole, 1 mM DTT). Cleavage and removal of the N-terminal Thio-His tag was achieved by incubation with 0.03 mg/ml TEV protease concomitant with dialysis into NTD dialysis buffer (20 mM Tris–Cl pH 8.0, 200 mM NaCl, 1 mM DTT), followed by a subsequent purification with the 5 ml Histrap HP column. TDP-43 NTD was further purified by SEC over a Superdex 75 26/60 column (GE Healthcare) with NTD SEC buffer (20 mM Tris–Cl pH 8.0, 50 mM NaCl, 1 mM DTT). Purified TDP-43 NTD was concentrated to ~500 μM, flash frozen, and stored at −80°C.

For purification of full-length TDP-43-MBP fusion protein, cell cultures were grown at 37°C to an OD600 of 0.6–0.9. Cell cultures

were then cooled to 16°C, after which protein expression was induced with addition of IPTG to 1 mM, followed by incubation at 16°C overnight. Cells were harvested by centrifugation, resuspended in TDP-43 binding buffer (20 mM Tris–Cl pH 8.0, 1 M NaCl, 10 mM imidazole, 10% (v/v) glycerol, 1 mM DTT) supplemented with complete EDTA-free protease inhibitor cocktail, and lysed with the EmulsiFlex C3. Cell lysates were clarified by centrifugation, applied to the 5 ml Histrap HP column, and washed with five volumes of TDP-43 binding buffer. TDP-43-MBP was eluted with a linear gradient of TDP-43 elution buffer (20 mM Tris–Cl pH 8.0, 1 M NaCl, 500 mM imidazole, 10% (v/v) glycerol, 1 mM DTT).

Eluant was further purified by SEC over a Superdex 200 26/60 column (GE Healthcare) with TDP-43 SEC buffer (20 mM Tris–Cl pH 8.0, 300 mM NaCl, 1 mM DTT). Purified TDP-43-MBP was concentrated to 250 μM, flash frozen, and stored at −80°C.

## NMR spectra processing

NMR spectra were processed with NMRPipe (Delaglio *et al*, 1995) and analyzed with Sparky (T. D. Goddard and D. G. Kneller, SPARKY 3, University of California, San Francisco).

Backbone amide resonance assignments for the C39S/C50S were determined using standard Bruker three-dimensional triple-resonance experiments: HNCA, HNCO, HN(CA)CO, CBCA(CO)NH, and HNCACB.

## NMR chemical shift deviations

$^{15}$N and $^{1}$H chemical shifts were measured from the crosspeaks on the HSQCs of 5, 20, 100, 200, 350 and 500 μM WT and 5, 100 and 200 μM variants in the condition of 20 mM HEPES pH 6.8, 1 mM DTT, and 10% $D_2O$. Chemical shift deviations for $^{1}$H and $^{15}$N were quantified by subtracting the chemical shift of the 5 μM sample from the respective value of each other samples. The average chemical shift deviations (CSDs) were calculated by the equation:

$$\Delta\delta 15N - 1H = \sqrt{\frac{1}{2}\left(\Delta\delta H^2 + (0.14\Delta\delta N)^2\right)}.$$

For the titrations of S48E and Y4R, the 100 μM of either $^{15}$N S48E was titrated by 50 μM, 100 μM, 400 μM, 1 mM, 1.6 mM, and 2 mM of natural isotopic abundance Y4R. $^{15}$N Y4R was titrated by 50 μM, 100 μM, 400 μM, 800 μM, 1 mM, and 2 mM of natural isotopic abundance S48E. The CSDs for $^{15}$N and $^{1}$H were quantified by subtracting the chemical shifts of 100 μM $^{15}$N S48E (or $^{15}$N Y4R) from the respective value of each mixture samples and then calculated by the equation above.

$^{1}$H-$^{15}$N HSQCs were acquired at 500 MHz $^{1}$H Larmor frequency with 32 transients per data point with 64* and 1,536* complex pairs in the indirect $^{15}$N and direct $^{1}$H dimensions, with corresponding acquisition times of 35 and 219 ms, and sweep widths of 36.0 and 14.0 ppm centered at 116.5 and 4.7 ppm, respectively.

## NMR paramagnetic relaxation enhancement

Since TDP-43 1–80 contains two intrinsic cysteines, C39 and C50, either cysteine was substituted by serine to generate the variants C39S and C50S, respectively. Other PRE variants including single

cysteine substitution at S2 and S29 were engineered from the construct of C39S C50S TDP-43 1–80. One mM MTSL (Toronto Research Chemicals, O875000) or 1 mM AcMTSL (diamagnetic analog of MTSL, Toronto Research Chemicals, A167900) was added into 50 μM of each variant protein under the condition of 20 mM Tris–Cl pH 8.0, 150 mM NaCl. After incubating 1 h at room temperature, the excess label was removed by HiPrep desalting column and the protein was exchanged to the buffer of 20 mM HEPES pH 6.8. Finally, the paramagnetic probe MTSL (or the diamagnetic acetylated MTSL analog, AcMTSL) was conjugated at C2, C29, C39, and C50, respectively. The samples were concentrated and flash frozen. For the intermolecular PRE measurement, 100 μM MTSL (or AcMTSL)-conjugated PRE variant was mixed with 100 μM $^{15}$N-labeled C39S/C50S in 20 mM HEPES pH 6.8, 10% $D_2O$.

Backbone amide proton transverse relaxation rate constant, $^{1}$H$_N$ $R_2$, were measured at 500 MHz $^{1}$H frequency for paramagnetic and diamagnetic samples, with 192* and 1,536* complex pairs in the $^{15}$N indirect and $^{1}$H direct dimensions, corresponding acquisition times of 105 and 21.9 ms, and sweep widths of 36 and 14 ppm centered at 116.5 and 4.7 ppm, respectively. Each $^{1}$H$_N$ $R_2$ experiment comprised three interleaved $^{1}$H$_N$ $R_2$ relaxation delays: 0.1, 2.1, 5.1 ms. PRE values ($\Gamma_2 = {}^{1}$H$_N$ $R_2^{para} - {}^{1}$H$_N$ $R_2^{dia}$) were obtained from the data collected with 0.1- and 2.1-ms delays using the two time point method as described previously (Iwahara *et al*, 2007).

## NMR structure calculation

NMR experiments were carried out at 25°C on Bruker Avance 500 or 850 MHz spectrometer on two separate samples of TDP-43 NTD dimer described below. Our first sample was made up of 0.7 mM $^{13}$C, $^{15}$N-labeled S48E 1–80 NTD mixed with 2 mM non-labeled Y4R, while our second sample was reciprocally prepared with labeled 0.7 mM $^{13}$C, $^{15}$N-labeled Y4R mutant mixed with 2 mM unlabeled S48E mutant. Both samples were prepared in identical buffer comprising of 20 mM HEPES pH 6.8 and 1 mM DTT. Each of these samples contains roughly 92% dimer species and facilitated NMR observation of single subunit of the dimer. Practically, complete NMR resonance assignments were attained using triple-resonance experiment performed using standard pulse programs available in Topspin 3.2 software (HNCA, HNCO, CBCACONH, HCCH-COSY, HCCH-TOCSY, HBHACONH, hbCBcgcdceHE, and hbCBcgcdHD). Distance restraints were derived from sets of 3D- NOESY experiments acquired with mixing time of 120 ms. Isotope-filtered NOESY experiments ($^{13}$C, $^{15}$N-filtered/edited 3D NOESY-$^{1}$H-$^{15}$N/$^{1}$H-$^{13}$C HSQC) were performed to identify intermolecular NOE, which were further classified in strong, medium and weak restrains corresponding to 3.5, 4.5, and 6.0 Å upper limits. Dihedral angle restraints were calculated by using software TALOS+ (Shen *et al*, 2009), while hydrogen bond restrains were identified from hydrogen–deuterium exchange and interpretation of NOESY patterns. Semi-automated NOESY assignment and initial structure calculation was performed using software CYANA (version 3.9) (Guntert, 2004), and final water refinement was carried using software Xplor-NIH (version 2.34) (Schwieters *et al*, 2003). The NMR assignments and structure coordinates for TDP-43 NTD dimer between S48E and Y4R mutants are deposited under BMRB entry number 30345, and PDB ID number 6B1G.

 

## Turbidity measurements

Turbidity was quantified by absorbance of 600 nm wavelength light for wild-type or S48E TDP-43-MBP diluted to 2.5 μM in 20 mM HEPES, 1 mM DTT, pH 7.0 with 150 mM, 300 mM, or 500 mM NaCl. Phase separation was initiated by addition of TEV protease to 0.003 mg/ml, and turbidity was monitored over a 12-h time period at 5-min intervals using a Spectra Max M5 microplate reader. All measurements were performed in triplicate.

## Microscopy of *in vitro* TDP-43 phase separation

Full-length TDP-43-MBP was diluted to 2.5 μM or 20 μM in 20 mM HEPES, 1 mM DTT, pH 7.0 supplemented with 150–300 mM NaCl. Phase separation was initiated by the addition of TEV protease to 0.006–0.030 mg/ml, and phase separation was monitored by DIC micrographs acquired at 30-min intervals over a 2-h time period. All images were acquired with a Zeiss Axiovert 200M with a 40× objective with a numerical aperture of 0.75. Images were processed with ImageJ.

## Modeling of the dimer interface

A molecular model for the TDP-43 NTD dimerization interface was generated with XPLOR-NIH (Schwieters *et al*, 2003). The published solution structure of the TDP-43 NTD (Mompean *et al*, 2016) (PBD ID: 2N4P), with the first 12 residues encompassing the His tag removed, was duplicated and randomly rotated a maximum of 360° and offset by 10–100 Å using the XPLOR-NIH python interface. This process was repeated 100 times in order to generate 100 randomized initial structures used for the IVM rigid body, torsion angle docking procedure (Schwieters & Clore, 2001). The dimerization interface was represented by highly ambiguous distance restraints derived from $^1$H-$^{15}$N chemical shift perturbations as described previously (Clore & Schwieters, 2003). Cutoffs of 0.02 and 0.01 ppm were used to define contact residues on the WT and S48E binding surfaces, respectively. Residues without solvent exposed side chains were excluded, as were C-terminal residues of S48E due to their disorder and relatively large distance from the interface. Relative orientation of the monomer subunits was restricted using ambiguous distance restraints derived from $^1$H$_N$ PRE data (Zwolak *et al*, 2010), with paramagnetic spin labels positions at S2, S29, C39, or C50, using a cutoff of > 10 s$^{-1}$. A hard square well-potential term with r$^{-6}$ sum averaging was used for both CSP-derived and PRE-derived distance restraints, with lower and upper cutoffs of 1.2–5 and 1.2–20 Å, respectively. A radius of gyration potential term corresponding to 2.2N$^{0.38}$–1, where N is the number of residues in the dimer (Kuszewski *et al*, 1999), was geometrically scaled over the course of the simulated annealing protocol to ensure compaction of the monomeric subunits in the dimer model. Each subunit was treated as a rigid body, with rotational and translational degrees of freedom. The lowest energy structure was chosen to represent the TDP-43 NTD dimer model.

## CG-MALS data collection and analysis

Light-scattering measurements for TDP-43 NTD were performed in a buffer containing 20 mM HEPES pH 6.8, 150 mM NaCl, and 1 mM DTT, for NTD WT also at 300 mM NaCl. Measurements were collected using a Calypso system (Wyatt Technology Corporation), which uses a software-controlled multiple syringe pump to create a concentration gradient, and a DAWN-HELEOS multi-angle light-scattering photometer to collect data from the incoming sample stream. Static light-scattering data were collected at 14 scattering angles as a function of protein concentration. For each injection, the solution was allowed to come to equilibrium for 60 s. Equilibrium MALS data as a function of concentration were analyzed as described by Attri *et al* (2010), and data obtained at all angles in a single experiment were combined for subsequent model generation. Simple monomer–dimer, monomer–trimer, etc., interactions and isodesmic self-association models were used to model the data. In a solution, in which the different scattering species X$_i$ correspond to different association states of a single protein, the theory of Rayleigh scattering from multicomponent solutions yields the concentration-dependent Rayleigh ratio $R$

$$\frac{R}{K^*} = \sum_i (iM_x)^2 [X_i],$$

in which $M_x$ is the molar mass of protein $X$, and $[X_i]$ is the concentration of the species $X_i$. $R$ is normalized to an optical constant $K^*$ defined as

$$K^* = \frac{4\pi^2 n_0^2}{N_A \lambda_0^4} \left(\frac{dn}{dc}\right)^2,$$

where $n_0$ denotes the refractive index of the solvent, $\lambda_0$ the vacuum wavelength of incident light (690 nm), $N_A$, Avogadro's number, and d$n$/d$c$ is the refractive index increment of the sample (a standard value for proteins of 0.185 was used). The concentrations of each species are related to the equilibrium constants and total protein concentration, resulting in the following equations for typical monomer–dimer, monomer–trimer, monomer–i-mer equilibria:

$$iX \rightleftharpoons X_i; K_A^{(i)} = \frac{[X_i]}{[K]^i}; [X]_{\text{total}} = \sum_i i[X_i].$$

Above, $i = 1$ represents the free monomer, the total molar concentration $[X]_{\text{total}}$ is known at each gradient injection, and $R(0)/K^*$ is measured. Non-linear least square optimization is used to obtain a single $K_A$ value that fits the data across the entire concentration range of interest.

To describe isodesmic self-association, we used equations previously described (Attri *et al*, 2010):

$$K_A = \frac{[X_i]}{[X_{i-1}][X_1]}; [X]_{\text{total}} = \sum_{i=1}^{\infty} i[X_i] = \frac{[X_1]}{(1 - K_A[X_1])^2}.$$

Fitting parameters were $K_A$ (i.e., 1/$K_D$) and the molecular weight of the building block, which was in agreement with a TDP-43 NTD monomer within error.

## SEC-MALS data collection and analysis

The SEC multi-angle light-scattering (SEC-MALS) experiments were carried out using a WTC-0150S5 (MW range 500–150,000 Da) size-exclusion column (Wyatt Technologies, Santa Barbara, CA, USA)

with three detectors connected in series: an Agilent 1200 ultraviolet (UV) detector (Agilent Technologies, Santa Clara, CA, USA), a Wyatt DAWN-HELEOS-II multi-angle light-scattering (MALS) detector, and a Wyatt Optilab T-rEX differential refractive index (RI) detector (Wyatt Technologies, Santa Barbara, CA, USA). The column was equilibrated with 20 mM HEPES pH 6.8, 1 mM DTT, and either 0 or 150 mM NaCl. All data were collected at 25°C. A 100 µl sample was injected into the column using an auto-sample injection method with a flow rate of 0.5 ml/min. Protein in the eluent was detected via UV absorbance at 280 nm, light-scattering, and refractive index detectors. The data were recorded and analyzed with the Wyatt Astra software (version 7.0).

### In vitro cross-linking reactions

Protein samples of TDP-43 NTD WT and S48E were prepared at 400 µM in the following buffer: 20 mM HEPES pH 7.5, 150 mM NaCl, and 1 mM DTT. The amine cross-linker BS3 (bis(sulfosuccinimidyl)suberate, Fisher Scientific) was added to create a molar ratio of 50. Reactions were incubated at room temperature for 30 min. The reactions were quenched by the addition of 100 mM Tris pH 7.5 and were incubated at room temperature for at least 15 min prior to taking samples for analysis on a gel.

### Analytical ultracentrifugation experiments

Sedimentation velocity experiments were conducted in a ProteomeLab XL-I analytical ultracentrifuge (Beckman Coulter, Indianapolis, IN, USA) following standard protocols unless mentioned otherwise (Zhao *et al*, 2013). The samples of TDP-43 NTD at 32.78, 98.83, and 295.60 µM concentration in a buffer containing 20 mM HEPES pH 6.8, 150 mM NaCl, and 1 mM DTT were loaded into a cell assembly comprised of a double-sector charcoal-filled centerpiece with a 12-mm path length and sapphire windows. Buffer density and viscosity were determined in a DMA 5,000 M density meter and an AMVn automated micro-viscometer (both Anton Paar, Graz, Austria), respectively. The partial specific volumes and the molecular masses of the proteins were calculated based on their amino acid compositions (Cohn & Edsall, 1943) in SEDFIT (https://sedfitsedphat.nibib.nih.gov/software/default.aspx). The cell assembly, containing identical sample and reference buffer volumes of 360 µl, was placed in a rotor and temperature equilibrated at rest at 20°C for 2 h before it was accelerated from 0 to 50,000 rpm. Rayleigh interference optical data were collected continuously for 12 h separately. The velocity data were modeled with diffusion-deconvoluted sedimentation coefficient distributions c(s) in SEDFIT, using algebraic noise decomposition and with signal-average frictional ratio and meniscus position refined with non-linear regression (Schuck, 2000, 2016). The s-values were corrected for time, and finite acceleration of the rotor was accounted for in the evaluation of Lamm equation solutions (Zhao *et al*, 2015). Maximum entropy regularization was applied at a confidence level of P-0.68.

Sedimentation equilibrium was attained at a rotor temperature of 20°C at increasing speeds of 12,000 (for 36 h), 25,000 (for 24 h), and 43,000 rpm (for 24 h) (Zhao *et al*, 2013). Protein at concentration of between 28.28 and 89.23 µM (130 µl) was loaded into double-sector centrepieces and absorbance distributions recorded at

280 in 0.001-cm-radial intervals with 20 replicates for each point. Global least squares modeling was performed at multiple rotor speeds with the software SEDPHAT (https://sedfitsedphat.nibib.nih.gov/software/default.aspx) using the reversible one-step self-association model. The association scheme used in this analysis was $[A] \leftrightarrow [A_2] \leftrightarrow [A_3] \leftrightarrow [A_4] \leftrightarrow \ldots$ [isodesmic] with A being the TDP-43 NTD monomer, and $K_D^{iso}$ the single dissociation constant for the association steps were obtained from the analysis. Errors of the fits represent the 68.3% confidence interval (CI) using an automated surface projection method (Zhao *et al*, 2015). All plots were generated with the program GUSSI (kindly provided by Dr. Chad Brautigam).

### Custom antibodies, HEK293T cell culture, and immunoblotting

Oligopeptides and phospho-specific rabbit antibodies against TDP-43 amino acids 40–53 (GLRYRNPVSQCMRG) were custom produced by GenScript (Piscataway, NJ USA). Confirmation of antibody specificity was performed by serially diluting TDP-43 oligopeptides in 8 M urea and spotting to nitrocellulose membranes, which were then blocked in 6% milk in Tris-buffered saline (TBS). Membranes were then probed with polyclonal antibodies specific to the phospho-peptide (α-TDP-43 pSer48) or non-specific to the phosphorylated state (PAN antibody). Primary antibodies were detected with IRDye conjugated secondary antibodies (LI-COR 926-32211 or 925-68020, Lincoln, NE, USA), and their fluorescence was measured and imaged using the Odyssey LI-COR Imaging System.

HEK293T cells were obtained from ATCC (CRL-11268) and maintained under standard conditions at 37°C in DMEM-based media (Dulbecco's modified Eagle's medium) supplemented with 10% fetal bovine serum and 1% P/S/G (penicillin–streptomycin–glutamine). Lysates were prepared by suspension and incubation of cells in nondenaturing lysis buffer (200 mM NaCl, 100 mM Tris–HCl, 0.5% sodium deoxycholate, 1% Triton X-100, 0.2% sodium dodecyl sulfate, 660 µM phenylmethylsulfonyl fluoride, 100 µl protease inhibitor cocktail (Sigma-Aldrich P8215, St. Louis, MO, USA) 100 µl phosphatase inhibitor cocktail (Sigma-Aldrich P0044), 1,250 units Benzonase nuclease (Sigma-Aldrich E1014).

Western blotting was performed using standard laboratory procedures with Mini-PROTEAN® Precast AnyKD™ acrylamide gels (Bio-Rad 456-9034) and nitrocellulose membranes (Bio-Rad 162-0146). Immunoblotting with custom and commercial α-TDP-43 antibodies (CST 3449) and detection with the LI-COR imaging system were performed as described above. Immunoprecipitation of HEK293T cell lysates was performed according to the manufacturer's protocol with Protein G Dynabeads (Invitrogen 10007D) and mouse monoclonal α-TDP-43 antibody (Proteintech 60019-2).

Phosphatase treatment of nitrocellulose membranes was performed using calf intestinal phosphatase (CIP; NEB M0290S). Following detection of the 43 kDa species by Western blotting, membranes were stripped of primary and secondary antibodies using stripping reagent (Thermo Fisher 46430). Membrane sections containing putative TDP-43 species were cut and blocked with 5% bovine serum albumin and 0.1% Triton X-100 in TBS. Identical membrane pieces were treated with and without CIP in reaction buffer (100 mM NaCl, 50 mM Tris–HCl, pH 8, 10 mM MgCl₂, 1 mM DTT) at 37°C. The membranes were then probed as described above.

## In cell phase separation assays

### DNA constructs

Plasmids used for this experiment are summarized below. Sequences and maps are available upon request. TDP43 was originally obtained from Addgene (ID 27462). Mammalian expression constructs were cloned into modified pCS-DEST backbones (Addgene 22423).

| Construct name | Plasmid | Construct description | Promoter | Used in figures |
|---|---|---|---|---|
| TDP-43 RRM-GFP | pHBS838 | TDP-43 with its RRM domains replaced by GFP | CMV | Fig 2C, E and F |
| TDP-43 RRM-GFP S48E | pHBS1006 | TDP-43 RRM-GFP with S48E mutation | CMV | Fig 2C, E and F |
| TDP-43 RRM-mCherry | pHBS853 | TDP-43 with its RRM domains replaced with mCherry | CMV | Fig 2C, E and F |

*Note:* Mammalian expression constructs used in this study.

### Cell culture

All TDP-43 phase separation reporter assays were performed in HEK-293T cells obtained from the American Tissue Culture Consortium (ATCC; Catalog Number CRL-3216, RRID:CVCL_0063). Cells were maintained and imaged at 37°C and 5% $CO_2$ in DMEM high glucose (GE Healthcare) supplemented with 10% FBS, 2 mM L-glutamine (Gibco), 1 mM non-essential amino acids (Gibco), and 1% penicillin/streptomycin (Gibco), unless otherwise noted.

### TDP-43 reporter phase formation

Mammalian TDP-43 expression constructs (500 ng) were transfected into HEK-293T cells grown on tissue culture-treated μ-slides (ibidi) with X-tremeGENE 9 (Roche). To compare expression levels, cells were harvested 24 h after transfection and analyzed by (i) flow cytometry with a BD Accuri C6 instrument (excitation laser: 488 nm, emission filter: 586/40 nm) or (ii) immunoblotting using anti-GFP (Novus NB600-308) and anti-alpha-tubulin (Sigma T6199) primary antibodies. Blots were developed with a LiCor Odyssey imager using appropriate dye-conjugated secondary antibodies. To follow TDP-43 reporter phase formation over time, cells were imaged in L15 medium (Gibco) + 10% FBS at 37°C every hour for a total of 24 h after transfection with a Leica DMI-6000B microscope equipped with an Andor Yokogawa spinning disk confocal unit. A 488-nm laser was used for GFP excitation and an Andor iXon Ultra EMCCD camera for signal detection.

### Quantification of TDP-43 reporter levels

A Leica SP8 confocal laser scanning microscope equipped with a 64× glycerol (NA 1.4) objective (Leica) and a temperature- and gas-controlled incubation chamber (Life Imaging Services) was used for image acquisition. The 488- and 561-nm laser lines were used for excitation of GFP (green channel) or mCherry (red channel), respectively. For quantification of TDP-43 RRM-GFP nuclear and droplet signals in live cells, each confocal image was taken with identical detector settings. Mathematica (Wolfram Research) was used to outline nuclei based on the total nuclear GFP signal, and to quantify the fluorescence signal of different TDP-43 RRM-GFP variants in the nucleoplasm versus the droplet phase.

### TDP-43 reporter phase dynamics

Half-bleach experiments were performed 24 h after transfection of the indicated constructs using the FRAP module of the Leica SP8 microscope. The Leica Application Suite X software was used for quantification and GraphPad Prism to plot and analyze replicate half-bleach experiments. For each recorded time point ($t$), the fluorescence intensities within the bleached droplet hemisphere were integrated and normalized to the fluorescence intensity of the corresponding unbleached droplet hemisphere. These normalized, time-dependent fluorescence intensities It were then used to calculate the fluorescence recovery (FR) according to the following formula: FR $(t) = (I_t - I_{t0})/(I_{before\ bleaching} - I_{t0})$, with $t0$ being the first time point observed after photobleaching.

## TDP-43 in cell splicing assays

### Materials

All reagents are from Sigma-Aldrich unless otherwise specified. HeLa cells were purchased from ATCC.

### Plasmid construction

FLAG-tagged wild-type TDP-43, siRNA-resistant construct for mammalian expression was previously described (4). This served as template for site-directed mutagenesis using the QuikChange Site-Directed Mutagenesis Kit (Agilent Technologies) to generate TDP-43 mutants for mammalian expression. Oligonucleotides used for mutagenesis: Y4R_FW, gacgacaagctttctgaacgtattcgggtaaccgaagatg; Y4R_RV, catcttcggttacccgaatacgttcagaaagcttgtcgtc; E17R_FW, agaacg atgagcccattcgaataccatcggaagacgatg; E17R_RV, catcgtcttccgatggtattcg aatgggctcatcgttct; S48A_FW, cgctacaggaatccagtggctcagtgtatgagaggtg tc; S48A_RV, gacacctctcatacactgagccactggattcctgtagcg; S48E_FW, cgctacaggaatccagtggaacagtgtatgagaggtgtc; S48E_RV, gacacctctcatacac tgttccactggattcctgtagcg.

### Cell culture, fractionation, and splicing assays

Human cell lines were grown in growth media-Dulbecco's modified Eagle's medium, 4,500 mg/l glucose, L-glutamine, and sodium bicarbonate and supplemented with filtered fetal bovine serum at 10%. RNAi-mediated downregulation of TDP-43 was carried out as previously described (1) and siCONTROL Nontargetting siRNA#1 (Dharmacon) was used as control. TDP-43 downregulation was confirmed by immunoblotting.

Splicing assays were carried out in HeLa cells transiently transfected with Lipofectamine 2000 and Oligofectamine in the case of siRNA (Life Technologies) according to manufacturer protocols. Splicing assays were performed using the CFTR reporter minigene as previously described (Li *et al*, 2017).

Immunoblotting for TDP-43 Western blot was carried out using approximately 30 μg of total protein. An antibody against TDP-43 (ProteinTech 10782-2-AP) was used to detect endogenous and tagged TDP-43. GAPDH was used as the loading control (Abcam ab181602).

**Expanded View** for this article is available online.

## Acknowledgements

We thank Geoff Williams at the Leduc Bioimaging Facility at Brown University for microscopy assistance. Research reported in this publication was supported in part by the National Institute of General Medical Sciences (NIGMS) of the National Institutes of Health (NIH) under Award Number R01GM118530 (to N.L.F), R35GM119790 (to F.S.), and R01GM112846 (T.M.), a subproject as part of an Institutional Development Award (IDeA) from NIGMS (P20GM104937), a starter grant 17-IIP-342 from the ALS Association (to N.L.F.), an ALS Research grant from the Judith & Jean Pape Adams Charitable Foundation (to N.L.F), and a grant from the American Lebanese Syrian Associated Charities (T.M.). A.E.C and V.H.R. were supported in part by training grants from NIGMS T32GM07601 and NIMH T32MH020068, respectively, and Brown Institute for Brain Science Graduate Awards. Work at Stanford was supported by grants from the NIH (DP2GM105448 and R35GM118082) to RR and a fellowship from the Deutsche Forschungsgemeinschaft (SCHM 3082/2-1) to HBS. This research is based in part on data obtained at the Brown University Structural Biology Core Facility supported by the Division of Biology and Medicine, Brown University. We thank Christoph Schorl and the Brown Genomics Core Facility supported by NIGMS P30GM103410, NCRR P30RR031153, P20RR018728, and S10RR02763, National Science Foundation EPSCoR 0554548. The content is solely the responsibility of the authors and does not necessarily represent the official views of the funding agencies.

## Author contributions

NLF, AW, AEC, DRM, MTN, and VHR designed and conducted the gel filtration, protein expression and purification, and structural NMR experiments. AEC developed the C-terminal MBP vector and phase separation assay. MTN solved the structure of the dimer from samples made by AW and data collected together with AW. NLF, AW, and AEC wrote the manuscript with input from the authors. TM and EWM conceived and analyzed the CG-MALS and cross-linking experiments, and EWM carried them out. TM, EWM, and AN conceived the AUC experiments, and AN carried out and analyzed them. HBS conceived, performed, and analyzed the phase separation cell culture experiments, with input and resources provided by RR. YMA and ANR conceived and analyzed the splicing experiments, and ANR carried them out. FS and SNR conceived of, performed, and analyzed the immunoblotting experiments.

## Conflict of interest

The authors declare that they have no conflict of interest.

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
