## [Review Process File · The EMBO Journal]

A single N-terminal phosphomimic disrupts TDP-43 polymerization, phase separation and RNA splicing

Ailin Wang, Alexander E. Conicella, Herman Broder Schmidt, Erik W. Martin, Shannon N Rhoads, Ashley N. Reeb, Amanda Nourse, Daniel Ramirez Montero, Veronica H. Ryan, Rajat Rohatgi, Frank Shewmaker, Mandar T. Naik, Tanja Mittag, Yuna M. Ayala, Nicolas L. Fawzi

Review timeline:

Submission date:	27 May 2017
Editorial Decision:	5 July 2017
Revision received:	21 October 2017
Accepted:	30 December 2017

Editor: Karin Dumstrei

Transaction Report:

1st Editorial Decision

5 July 2017

Thank you for submitting your manuscript to The EMBO Journal. Your study has now been seen by 2 referees and their comments are provided below.

As you can see from the comments below both referees find the analysis interesting, but referee #1 also raises a number of significant concerns. One of the main concerns raised by referee #1 is that there is no data on the phosphorylation of the S48 site and if this is happening in normal or pathological conditions. I agree with the referee that such insight would be nice to have, but also find that this is beyond the scope of this analysis. I do see the value of analysing the S48 TDP-43 phosphomimetic as this gives you a handle on understanding the functional consequences of the ability of the TDP-43 N-terminal domain to form head-to-tail linear chains. A number of other concerns are also raised, but some of them should be fairly easy to resolve.

Given the comments provided, I would like to invite you to submit a revised manuscript. As stated above, we don't need any further insight into the phosphorylation of S48 or its regulation, but please make sure that you have a balanced discussion regarding potential physiological/pathological significance. Regarding the other comments raised, I find many of them very relevant and would like to discuss with you what can be done in a timely fashion to address these. Please send me a point-by-point response and we can then discuss further.

When preparing your letter of response to the referees' comments, please bear in mind that this will form part of the Review Process File, and will therefore be available online to the community. For more details on our Transparent Editorial Process, please visit our website: http://emboj.embopress.org/about#Transparent_Process

We generally allow three months as standard revision time. As a matter of policy, competing manuscripts published during this period will not negatively impact on our assessment of the conceptual advance presented by your study. However, we request that you contact the editor as soon as possible upon publication of any related work, to discuss how to proceed. Should you foresee a problem in meeting this three-month deadline, please let us know in advance and we may

be able to grant an extension.

Thank you for the opportunity to consider your work for publication. I look forward to your revision.

REFEREE REPORTS

Referee #1:

Wang and co-workers investigate the effect of a phosphomimic mutation in TDP-43 N-terminal domain, providing observations on the importance of S48 residue in protein polymerization and functionality.

This work deals with key aspects of TDP-43 molecules interactions which could have a high impact in better understanding the nature of protein structure, interactions and function both in physiological and in pathological contexts. However, the manuscript appears highly speculative with respect to the role of pS48. Particularly because beyond the "phosphoproteomic screens" mentioned in this paper there is no back up data on the presence of functional S48 phosphorylation in normal and/or pathological tissues. Further experiments need to be done in order to confirm some of the stated hypotheses.

Specific points are as follows:

1:

Page 3, line 2: On the basis of the phosphoproteomic screens (Hornbeck et al, 2015; Rigbolt et al, 2011) the authors stated that S48 is a "known post translational modification site", but until today, nothing is known about this pathological hyperphosphorylation in vivo. It seems excessive to base the work and the conclusions on a phosphorylated site that has not been quantified and really validated in the cellular context.

2:

Pag.13: As the authors suggest at the end of discussion, a control of a S48 hyperphosphorylation assay would be helpful to further confirm the correlation between this pathological modification and TDP-43 LOF, also regarding the CFTR exon 9 splicing assay.

3:

Pag 13, line 16: The authors speculate about a cooperation between NTD and CTD for TDP-43 polymerization but this point has not been directly investigated and proved in this paper. Do the authors base their theory only on data obtained from Conicella's paper of 2016? Could the authors demonstrate that there is a direct double interaction between NTD and CTD? Would it be possible to check the effect on polymerization following mutations in NTD and CTD in full lenght TDP-43 WT and S48E mutants?

4:

Fig. 2I: It would be useful if the authors could add a gel showing CFTR exon 9 splicing pattern

5:

Fig. 2I: As shown by the authors, in the CFTR exon 9 splicing assay, the mutant S48A (which is phosphorylation deficient) also significantly decreases splicing activity. It therefore seems that disruption of polymerization would not necessarily be due to the hyperphosphorylation process itself in position 48, but rather to any change in that position. The authors should pay attention regarding speculation on the pathological role of S48 hyperphosphorylation. The NTD of TDP 43 has a very well defined structure whose disruption by point mutations causes loss of function independently of eventual phosphorylation.

Furthermore, in figure S4B it is shown that E17R and Y4R mutations also disrupt self-assembly in a way that appears to me even stronger than the S48E mutant.

Could the authors show CFTR exon 9 splicing regulation also in the case of E17R and Y4R mutants?

6:

Pag 8 last 2 lines: The sentence is not clear and disconnected, must be rephrased.

7:

Pag.11, line 28: Given that the authors proved that S48E and Y4R mutants are able to dimerize and preserve head-to-tail interface, it would be interesting to check if a co-transfection of these 2 mutants in siTDP HeLa cells, is able to recover TDP-43 functionality in CFTR exon 9 splicing.

Minor concerns:

Pag.2 "Hyperphosphorylation and ubiquitination of the deposited C-terminal fragments of TDP-43 in ALS patient cells and in animal and cell culture models, ... although the mechanistic link between CTD aggregation and cellular toxicity remains unclear": The sentence is disconnected and must be rephrased.

Pag.8: Last three lines: The sentence is not clear.

Referee #2:

The paper by Wang/Conicella and co-workers concerns the protein TDP-43. The authors show the protein can form liquid droplets both in vitro and inside cells and that a phosphomimicking mutation S48E affects this process. The authors further show that the folded NTD can form oligomers via specific head-to-tail interactions. Once formed, this interface occludes position 48. This provides a structural rationale for how modification of this site can effect function of this protein.

A range of biophysical techniques are employed, notably solution-state NMR to obtain a model for the NTD dimer and cell based assays for characterising phase separation and liquid droplet formation of the full length protein.

I find the work interesting, and it adds to exciting and rapidly expanding field focused on phase separating proteins. As such, I recommend publication in the EMBO journal.

I have a small number of technical queries for consideration by the authors. I found aspects of the article to be difficult to navigate. The figures and the controls should be linked up in the text and figure legends. I would also recommend more work going into the figure legends so that relevant experimental details are all there. I've no doubt at all that these concerns can be addressed easily.

1) I find the cell based phase separation data to be over-interpreted. Initially, it took me a while to find fig S2 that appears to have a control for protein concentration in the cell expression levels. This important figure does not appear to be referenced in the text but is important. Qualitatively, S48E appears to have lower overall expression, judging from the gel in S2. In principle, lower expression of S48E can explain the difference in the relative fluorescence inside and outside the droplets in the cell (Fig 2). Less protein puts WT and S48A in difference places in the phase diagram, so even if you had otherwise identical interaction strength, you'd expect more protein outside of the S48E droplets relative to WT.

So qualitatively, the finding that the relative fluorescence inside and outside of the droplets does not report on there being a difference in the interaction strength between molecules inside the liquid droplets. The authors should either quantify what they have more carefully to see if it holds quantitative information about interaction strengths, or refrain from interpreting this as a significant difference between the two proteins, if it can be explained by something trivial like expression levels.

The other two differences: that the mutant slows down internal diffusion (FRAP) and there are functional differences between WT and mutant both appear both nice and compelling.

2) The in cell phase separation data shows that S48E and WT can phase separate and form liquid droplets. Under the in-vitro conditions shown however, S48E did not phase separate. So there's a lack of agreement between the in vivo and the in vitro data that should be addressed in more detail.

Perhaps the authors concentration estimate in the in vitro experiments was too low to reflect what they have in cells, or cofactors were lacking in the in vitro experiments. Moreover, could the presence of the GFP tag be the difference between the in vivo and the in vitro experiments?

An easy way to get a thermodynamic measurement of the stability of droplets in cells would be to heat shock them. If S48E droplets disappear at a lower temperature than WT, then that would make a convincing demonstration that the stability of the liquid droplets inside cells is substantially different inside cells.

3) Fig 2c/d. The authors say nuclear droplets are observed after overnight expression. There appears to be a single droplet in the images shown? Presumably at early stages, a range of puncta with variable sizes are seen? Is the time taken from induction to onset of droplets similar for the wild type and mutant? The fluorescence within the bodies does not appear to be uniform. Why is this? From the data presented, apparent from internal diffusion, the droplets formed from WT and mutant appear more similar than different. Certainly the effects of the mutation is not large in the in-cell experiments.

4) The CTD and wild type liquid droplets formed in vitro harden over time:

Interestingly, as observed for the C-terminal domain alone (Conicella et al., 2016), after two-hour incubation these assemblies appear irregularly shaped, no longer flowing and fusing, consistent with aggregation within the LLPS state (Figure 2B).

Is this due to disulphide formation? Or by contrast, is there any evidence for amyloid formation (fibrils seen by EM on dilution or enhanced ThT fluorescence?).

5) The authors note that

Higher-order assembly is similar at the two different ionic strength conditions, suggesting that electrostatic interactions are not primarily responsible for stabilizing the interactions.

But at high salt, liquid droplet formation in vitro is inhibited. This, taken with the CTR forming liquid droplets on its own, and the similarities between the droplets formed in cells (Fig2) suggest that the CTR is providing the dominant forces for phase separation with the NTD providing only a relatively modest correction. A point to this effect should be made in the text: presently the impression left is that the two domains contribute similarly.

6) The authors talk of the NTD forming multivalent interactions. Each molecule provides 1 head and 1 tail interface according to the structural data. This is not multivalent nor cooperative (if the NTD interactions were cooperative, a uniform linear polymerisation model would not explain the data). In the discussion the authors are careful to distinguish between multivalency in the CTR and NTR:

we demonstrate that full-length TDP-43 also self-assembles into LLPS droplets via multivalent cooperation between the C-terminal domain and linear polymerization of the globular N-terminal domain.

But in the results, they are not so clear. So statements like these should be adjusted to reflect this:

NTD polymerization enhances phase separation of full-length TDP-43 by providing additional multivalent interaction sites creating cooperativity between NTD self-interactions consistent with reduced multivalent contacts
wild-type NTD normally engages in multivalent interactions

7) When talking about the effects of chemical exchange in NMR spectra, I would imagine most readers, particularly those without an NMR background would be baffled by the following. I would recommend clearly defining the effects expected for a more general readership.

the line broadening is reduced
Quenching of the line broadening

challenging due to exchange broadening

1st Revision - authors' response

21 October 2017

Referee #1:

Wang and co-workers investigate the effect of a phosphomimic mutation in TDP-43 Nterminal domain, providing observations on the importance of S48 residue in protein polymerization and functionality.

This work deals with key aspects of TDP-43 molecules interactions which could have a high impact in better understanding the nature of protein structure, interactions and function both in physiological and in pathological contexts. However, the manuscript appears highly speculative with respect to the role of pS48.

The manuscript as written was indeed initially speculative regarding pS48. S48E is used as a model that disrupts assembly and function of TDP-43. We did not intend to claim phosphorylation we have shown that is pathological or physiological functional - and experiments probing these questions are a related line of research that are exciting. However our feeling is that determining the structure of TDP-43 assemblies and linking to phase separation is extremely exciting and unites the work in the field on TDP-43. We have therefore carefully changed the wording of the interpretation of results and in the discussion to clarify this point. We have also added additional data supporting the constitutive phosphorylation of TDP-43 at low levels by a novel antibody (see below), consistent with previous phosphoproteomic results previously deposited in Phosphosite.

Particularly because beyond the "phosphoproteomic screens" mentioned in this paper there is no back up data on the presence of functional S48 phosphorylation in normal and/or pathological tissues. Further experiments need to be done in order to confirm some of the stated hypotheses.

We have changed the wording from "screens" to "analyses on several cell lines" to be more specific.

We have now added new experiments where we designed, tested, and used a custom polyclonal antibody serum specific for phosphoserine at S48 position in TDP-43. This antibody reacts with a band at 43kDa in HEK293T cell lysates. The band is eliminated upon phosphatase treatment of the membrane prior to antibody probing. A band at 43kDa when probed with the phosphospecific antibody is also present after IP with commercial anti-TDP-43 antibody (and reactivity is also eliminated with phosphatase treatment of the membrane), suggesting this band corresponds to pS48 TDP-43.

In summary, we feel the data on TDP-43 assembly via the N-terminal domain into phase separated assemblies is independently strong and that we have now provided sufficient new data supporting the presence of S48 phosphorylation in low but measurable amounts in human cell lines lysates and hence the potential relevance of phosphorylation of TDP-43 at S48. This also corroborates the multiple proteomic studies using human cells, which identified S48 as a phosphorylation site. Even if only a minority of TDP-43 is physiologically or pathologically phosphorylated, it may be a useful target to modulate assembly and hence function of TDP-43 for engineered or pharmacological reasons. We have updated the text to carefully qualify the relevance of S48 phosphorylation.

Specific points are as follows:

1: Page 3, line 2: On the basis of the phosphoproteomic screens (Hornbeck et al, 2015; Rigbolt et al, 2011) the authors stated that S48 is a "known post translational modification site", but until today, nothing is known about this pathological hyperphosphorylation in vivo. It seems excessive to base the work and the conclusions on a phosphorylated site that has not been quantified and really validated in the cellular context.

We would like to clarify - we do not believe TDP-43 pS48 is hyperphosphorylated in disease. This region is not included in the C-terminal fragments deposited in disease so it is not clear how it is connected to disease and we do not make that claim in the paper. It is a previously identified phosphorylation site that we have now provided additional evidence may be phosphorylated in cells, though we do not claim to know the function or potential regulatory mechanisms. We do not claim to know the kinase that deposits this phosphorylation. These are exciting avenues for future studies.

2: Pag.13: As the authors suggest at the end of discussion, a control of a S48 hyperphosphorylation assay would be helpful to further confirm the correlation between this pathological modification and TDP-43 LOF, also regarding the CFTR exon 9 splicing assay.

We would like to clarify - we do not claim phosphorylation at S48 is a pathological modification. In the future, it will be interesting to test the effect of protein inclusion formation and toxicity of the S48E and other NTD disrupting variants. A very recent report from Polymenidou and colleagues (Afroz et al Nature Communications 2017) showed in one cell type that transfected GFP-tagged wild type TDP-43 was less likely to form cytoplasmic C-terminal TDP-43 inclusions associated with ALS than GFP-tagged interface disrupting variants. This result suggests future work in stable cell line and organismal models of ALS would be interesting and further cell work without fusion tags is necessary to confirm the result. However, this is not the topic of our work here. Here we focus on the detailed structural biology of TPD-43 assembly mechanism (which we find is a specific kind of linear polymerization) and its effects on TDP-43 phase separation and physiological splicing function – we do not describe pathological assembly.

3: Pag 13, line 16: The authors speculate about a cooperation between NTD and CTD for TDP-43 polymerization but this point has not been directly investigated and proved in this paper. Do the authors base their theory only on data obtained from Conicella's paper of 2016? Could the authors demonstrate that there is a direct double interaction between NTD and CTD? Would it be possible to check the effect on polymerization following mutations in NTD and CTD in full length TDP-43 WT and S48E mutants?

We appreciate the reviewer's concern and do feel that clarification suggested is helpful and so we have made clear that NTD modulates phase separation mediated primarily by the C-terminal domain. (see Reviewer 2 comments).

We do suggest that the data we show here combined with our previous data (Conicella et al) effectively demonstrate that N-terminal domain interactions with other N-terminal domains cooperate with C-terminal domain interactions with other C-terminal domains to enhance phase separation of full length TDP-43. TDP-43 CTD is competent for phase separation alone at >5uM quantities in vitro (Conicella et al). Here TDP-43 full-length phase separates in vitro at <2.5uM quantities, but only with wild type and not S48E. NTD alone is not competent for phase separation (this paper – this experiment essentially serves as a “delta CTD” experiment). The final figure attempts to draw the complex nature of the interactions possible - it is not immediately clear how double interactions can be visualized as each chain may make contacts with distinct chains via the N- and C-terminal domains. We also show that full-length S48E has a different gel filtration profile (due to lack of assembly) than wild type. We have here presented the effect of NTD variant S48E on phase separation. We have not presented the effect of CTD variants on phase separation as this is outside the scope of this manuscript – we investigated CTD variants in depth in our previous work on the CTD alone (Conicella et al).

4: Fig. 2I: It would be useful if the authors could add a gel showing CFTR exon 9 splicing pattern

We thank the reviewer for this suggestion. The gel is now shown in Figure S2G.

5: Fig. 2I: As shown by the authors, in the CFTR exon 9 splicing assay, the mutant S48A (which is phosphorylation deficient) also significantly decreases splicing activity. It therefore seems that disruption of polymerization would not necessarily be due to the hyperphosphorylation process itself in position 48, but rather to any change in that position.

We agree with the reviewer and we have clarified this point. Indeed any chemical change at the site could alter interaction. In fact, we show that S48A disrupts interaction in vitro of the NTD. As S48 is near the interface, any change in the residue could modify interaction. We feel it is important to note as we do in the manuscript that phosphorylation at S48 could indeed cause this change in affinity by introducing steric bulk and like charge repulsion, and hence alter function.

The authors should pay attention regarding speculation on the pathological role of S48 hyperphosphorylation. The NTD of TDP 43 has a very well defined structure whose disruption by point mutations causes loss of function independently of eventual phosphorylation.

We agree with the reviewer that there are other ways to disrupt the NTD (including deletion of the

entire domain) and disrupt function. Several other groups have now proposed engineered variants that disrupt NTD interactions (some of them claim it is a dimer, tetramer, etc but our data show it is a linear polymer with head to tail assembly). Here we provide a possible modification that cells actually make. We have now carefully qualified the conclusions regarding the physiological or pathological significance of phosphorylation and have also provided additional evidence for phosphorylation. In any case, our observations that a phosphosite could regulate assembly is unique in our manuscript, as is the connection of function to phase separation in vitro and in cells.

Furthermore, in figure S4B it is shown that E17R and Y4R mutations also disrupt selfassembly in a way that appears to me even stronger than the S48E mutant.

Could the authors show CFTR exon 9 splicing regulation also in the case of E17R and Y4R mutants?

As suggested by the reviewer, we analyzed the splicing regulatory activity of Y4R and E17R in our cellular splicing assays using the CFTR exon 9 minigene reporter. The results are reported in Figure 2I showing that, as predicted, these substitutions greatly affect TDP-43 splicing activity. These new observations further support the role of NTD-mediated oligomerization in splicing regulation. Indeed, the degree of disruption to splicing appears to correlate with the disruption of assembly observed by gel filtration (Figure S4B) – however our gel filtration results were not performed quantitatively (they were preparative scale) so we do not feel it is appropriate to discuss that in the manuscript.

6: Pag 8 last 2 lines: The sentence is not clear and disconnected, must be rephrased.

We thank the reviewer for this suggestion and have now rephrased.

7: Pag.11, line 28: Given that the authors proved that S48E and Y4R mutants are able to dimerize and preserve head-to-tail interface, it would be interesting to check if a cotransfection of these 2 mutants in siTDP HeLa cells, is able to recover TDP-43 functionality in CFTR exon 9 splicing.

We thank the reviewer for this interesting suggestion. To attempt to address this question, mutants S48E and Y4R were co-overexpressed in siRNA treated HeLa cells. Co-expression of the two vectors did not show significant increase in TDP-43 activity compared to S48E alone. This may be due to uneven levels of mutant expression in co-transfected cells or toxicity resulting from higher levels of TDP-43 expression than in the standard well established and documented assay. Further experiments could test a range of vector concentrations to find conditions that produce similar levels of mutant expression, but we concluded they are out of the scope. Additionally, a negative result would not convincingly show that an NTD dimer is the relevant species for splicing function as the population of dimer would also be affected by decrease in the total number of competent interfaces even given equivalent amount of total TDP-43 compared to wild type control.

Minor concerns:

Pag.2 "Hyperphosphorylation and ubiquitination of the deposited C-terminal fragments of TDP-43 in ALS patient cells and in animal and cell culture models, ... although the mechanistic link between CTD aggregation and cellular toxicity remains unclear": The sentence is disconnected and must be rephrased.

We have rephrased this section.

Pag.8: Last three lines: The sentence is not clear.

We rephrased (we believe this is the same as question 6 “Pag 8 last 2 lines: The sentence is not clear and disconnected, must be rephrased.”)

Referee #2:

The paper by Wang/Conicella and co-workers concerns the protein TDP-43. The authors show the protein can form liquid droplets both in vitro and inside cells and that a phosphomimicking mutation S48E affects this process. The authors further show that the folded NTD can form oligomers via specific head-to-tail interactions. Once formed, this interface occludes position 48. This provides a structural rationale for how modification of this site can effect function of this protein.

A range of biophysical techniques are employed, notably solution-state NMR to obtain a model for the NTD dimer and cell based assays for characterising phase separation and liquid droplet formation of the full length protein.

I find the work interesting, and it adds to exciting and rapidly expanding field focused on phase separating proteins. As such, I recommend publication in the EMBO journal.

I have a small number of technical queries for consideration by the authors. I found aspects of the article to be difficult to navigate. The figures and the controls should be linked up in the text and figure legends. I would also recommend more work going into the figure legends so that relevant experimental details are all there. I've no doubt at all that these concerns can be addressed easily.

We thank the reviewer for these suggestions and have now implemented them throughout the manuscript.

1) I find the cell based phase separation data to be over-interpreted. Initially, it took me a while to find fig S2 that appears to have a control for protein concentration in the cell expression levels. This important figure does not appear to be referenced in the text but is important. Qualitatively, S48E appears to have lower overall expression, judging from the gel in S2. In principle, lower expression of S48E can explain the difference in the relative fluorescence inside and outside the droplets in the cell (Fig 2). Less protein puts WT and S48A in difference places in the phase diagram, so even if you had otherwise identical interaction strength, you'd expect more protein outside of the S48E droplets relative to WT. So qualitatively, the finding that the relative fluorescence inside and outside of the droplets does not report on there being a difference in the interaction strength between molecules inside the liquid droplets. The authors should either quantify what they have more carefully to see if it holds quantitative information about interaction strengths, or refrain from interpreting this as a significant difference between the two proteins, if it can be explained by something trivial like expression levels.

We thank Reviewer 2 for raising this important issue and apologize for any confusion regarding figure layout. We want to point out that original Fig. S2C shows controls for the splicing assay, but not for the phase separation assay. To show that the total amounts of WT and S48E in cells are indeed comparable, but the relative fractions of phase-separated and soluble protein differ, we corroborated our findings by western blotting (Fig. 2D and Fig. S2C), flow cytometry (Fig. S2D) and image quantification (Fig. S2E). In order to better convey this information, we have updated Figure 2 as follows:

- The original confocal images in panel C, which were taken with detector settings optimized to the different dynamic ranges of WT and S48E droplets, have been replaced by representative confocal images acquired at identical detector settings at two different sensitivities.
- Panel D has been replaced with an immunoblot indicating that total expression levels of WT and S48E TDP43RRM-GFP are comparable.
- The original figure of exon inclusion levels in panel I is replaced with the relative splicing activity of S48A and S48E, as well as the newly added Y4R and E17R. The original panel J is deleted.

In addition, Figure S2 has been amended with the following information:

- Panel C shows the full immunoblot of new Fig. 2D including markers.
- Determination of total WT and S48 protein levels by flow cytometry, also indicating that total expression levels of WT and S48E TDP43RRM-GFP are comparable. (Figure S2D)
- Quantification and statistical comparison of nuclear and droplet GFP signals for images acquired as in new Fig. 2C, indicating that the observed differences in nuclear and droplets signals are not large but are measurable and significant. (Figure S2E)
- Representative images of WT and S48E droplet formation over time showing differences in droplet formation kinetics. (Figure S2F)

Given that the overall concentrations of WT and S48E appear equivalent in our experiments and that the cells were kept at identical growth conditions (37°C), one can conclude that the interaction strength between S48E-S48E molecules is weaker than between WT-WT molecules. We want to

point out that additional experiments suggested by Reviewer 2 further strengthen this interpretation.

2) The in-cell phase separation data shows that S48E and WT can phase separate and form liquid droplets. Under the in-vitro conditions shown however, S48E did not phase separate. So there's a lack of agreement between the in vivo and the in vitro data that should be addressed in more detail.

Perhaps the authors concentration estimate in the in vitro experiments was too low to reflect what they have in cells, or cofactors were lacking in the in vitro experiments. Moreover, could the presence of the GFP tag be the difference between the in vivo and the in vitro experiments?

We appreciate this important suggestion. Previously, the critical concentration for TDP43RRM-GFP phase separation in cells was approximated to be $5 \mu\text{M} \pm 2.5 \mu\text{M}$ (Schmidt and Rohatgi, 2016). Based on these numbers, we estimate the critical concentration for the S48E mutant to be around 8-10 μM . Hence, the in vitro experiments were carried out at the lower end of the concentration range where one expects WT phase separation. We have now added additional in vitro experiments on full-length TDP-43 at higher concentrations showing that both WT and S48E can phase separate when the concentration is sufficiently high (in this case 20 μM in 300 mM NaCl buffer). Substitution of GFP in the in cell phase separation reporter assays for the RNA binding domains in the full length native protein (used in vitro) could alter the behavior. However, it should be stressed that there are many differences between in cell and in vitro (RNA, buffer/salts, other proteins and molecular components, chromatin). Hence our new data showing S48E is still competent for phase separation and our previous data showing the CTD alone is competent for phase separation alone provide the important requested correspondence between in vitro and in cell.

An easy way to get a thermodynamic measurement of the stability of droplets in cells would be to heat shock them. If S48E droplets disappear at a lower temperature than WT, then that would make a convincing demonstration that the stability of the liquid droplets inside cells is substantially different inside cells.

We agree with Reviewer 2 that phase transitions are often driven by temperature changes and that this would hence be an informative experiment in principle. However, because the droplet phases are highly stable, we find that the temperatures at which we observe differences in WT and S48E droplet stability are beyond what cells can sustain without gross changes in cell behavior, hence limiting our ability to obtain robust data. Even without this data, we nevertheless think that our findings sufficiently support the notion that interaction strengths are weaker for S48E than for WT TDP-43RRM-GFP in cells.

3) Fig 2c/d. The authors say nuclear droplets are observed after overnight expression. There appears to be a single droplet in the images shown? Presumably at early stages, a range of puncta with variable sizes are seen? Is the time taken from induction to onset of droplets similar for the wild type and mutant?

Indeed, when following droplet formation over time, we initially observe multiple puncta at early time points (Fig. S2F). It appears as if these eventually converge into single large droplets by Ostwald ripening and/or fusion. In agreement with our interpretation that the interaction strength between S48E mutants is weaker compared to WT molecules, we observe that the time it takes for puncta/droplets to form (after an initial nuclear signal is observed) is much greater for S48E than for WT (Fig. S2F) in representative examples presented in the paper.

The fluorescence within the bodies does not appear to be uniform. Why is this? From the data presented, apparent from internal diffusion, the droplets formed from WT and mutant appear more similar than different. Certainly the effects of the mutation is not large in the incell experiments.

We agree with reviewer 2 that the internal morphology of the TDP43 droplets is striking. As we previously described and discussed, the internal 'bubbles' or voids are regions filled with bulk nucleoplasm that are reminiscent of nucleolar vacuoles (Schmidt and Rohatgi, 2016). The presence of voids depends on the dynamic nature of the droplets. We agree with the reviewer –

apart from the diffusion the WT and S48E are similar, but not identical as we have now quantified.

4) The CTD and wild type liquid droplets formed in vitro harden over time:

Interestingly, as observed for the C-terminal domain alone (Conicella et al., 2016), after twohour incubation these assemblies appear irregularly shaped, no longer flowing and fusing, consistent with aggregation within the LLPS state (Figure 2B). Is this due to disulphide formation? Or by contrast, is there any evidence for amyloid formation (fibrils seen by EM on dilution or enhanced ThT fluorescence?).

We do not believe this is due to disulfide formation 1) because DTT is included in all experiments with native cysteine residues (i.e. all except PRE experiments where cysteine are substituted and/or used to attach the nitroxide radical via a disulfide) 2) because we extensively described the conversion of liquid to gel-like solid-like structures of the C-terminal domain alone that does not contain any cysteine residues. Full length TDP-43 does contain C39 and C50 in N-terminal domain which are involved in the interfaces (Figure 1H and Figure S3B) and do not in that context form a disulfide bond.

At this point, we have not observed evidence for fibrilization – morphologically we do not see fibrous aggregates as we have for FUS full-length (Monahan and Ryan et al.). It will be interesting to examine these states with TEM and other high resolution techniques, but we believe this is out of the scope.

5) The authors note that “Higher-order assembly is similar at the two different ionic strength conditions, suggesting that electrostatic interactions are not primarily responsible for stabilizing the interactions. “

But at high salt, liquid droplet formation in vitro is inhibited. This, taken with the CTR forming liquid droplets on its own, and the similarities between the droplets formed in cells (Fig2) suggest that the CTR is providing the dominant forces for phase separation with the NTD providing only a relatively modest correction. A point to this effect should be made in the text: presently the impression left is that the two domains contribute similarly.

We agree with the reviewer. Indeed CTD is able to phase separate on its own (Conicella et al). Therefore we have updated the text to reflect the dominant role of the CTD in phase separation.

6) The authors talk of the NTD forming multivalent interactions. Each molecule provides 1 head and 1 tail interface according to the structural data. This is not multivalent nor cooperative (if the NTD interactions were cooperative, a uniform linear polymerisation model would not explain the data). In the discussion the authors are careful to distinguish between multivalency in the CTR and NTR:

we demonstrate that full-length TDP-43 also self-assembles into LLPS droplets via multivalent cooperation between the C-terminal domain and linear polymerization of the globular N-terminal domain.

But in the results, they are not so clear. So statements like these should be adjusted to reflect this:

- ***NTD polymerization enhances phase separation of full-length TDP-43 by providing additional multivalent interaction sites creating cooperativity between NTD selfinteractions***
- ***consistent with reduced multivalent contacts***
- ***wild-type NTD normally engages in multivalent interactions***

We thank the reviewer for the suggestion and have altered the wording in these areas.

7) When talking about the effects of chemical exchange in NMR spectra, I would imagine most readers, particularly those without an NMR background would be baffled by the following. I would recommend clearly defining the effects expected for a more general readership.

- ***the line broadening is reduced***
- ***Quenching of the line broadening***
- ***challenging due to exchange broadening***

We have included text clarifying and altering these statements that are indeed NMR jargon.

In addition to the changes described here, we have strengthened the manuscript by providing the solution structure of the asymmetric dimer we described in the previous version (S48E + Y4R) and compared this structure to the previously available and newly available structures for TDP-43 NTD monomers and polymers. We have also commented on the significance of our results in light of

these recent findings. Our result remains the only one describing TDP-43 NTD connection to phase separation (which we show in cells and in vitro) and the phase separation assay on its own is a valuable contribution to the literature (the difficulty and importance was alluded to the failure of previous attempts by J Paul Taylor and coworkers to purify soluble TDP-43 full length is described in Molliex et al, *Cell* 2016) which we have expanded by demonstrating the behavior of the S48E variant in modulating (decreasing) phase separation is overcome by higher concentration of protein in vitro.

Accepted

30 December 2017

Thank you for submitting your revised manuscript to The EMBO Journal.

I am very sorry for the delay in getting back to you with a decision. As I have discussed with you referee #2 had agreed to re-review the manuscript but didn't deliver a referee report despite many calls and emails. I decided to seek further input on the study from a trusted advisor who looked at the manuscript, the referee comments and your revisions. I have now hear back from the advisor who finds that you have done a very good job in responding to the concerns raised and support publication here.

I am therefore very pleased to accept your manuscript for publication here.

Corresponding Author Name: Nicolas Fawzi
 Journal Submitted to: EMBO Journal
 Manuscript Number: EMBOJ-2017-97452R